# DreamFuser: Value-guided Diffusion Policy for Offline Reinforcement Learning

## Abstract

Recent advances in reinforcement learning have underscored the potential of diffusion models, particularly in the context of policy learning. While earlier applications were predominantly focused on single-timestep settings, trajectory-based diffusion policy learning promises significant superiority, especially for low-level control tasks. In this context, we introduce DreamFuser, a trajectory-based value optimization approach that seamlessly blends the merits of diffusion-based trajectory learning and efficient Q function learning over state and noisy action. To address the computational challenges associated with action sampling of diffusion policy during the training phase, we design the DreamFuser based on the Generalized Noisy Action Markov Decision Process (GNMDP), which views the diffusion denoising process as part of the MDP transition. Through the empirical tests, the expressive and optimization abilities of the DreamFuser are verified. The experiment results also reveal DreamFuser's advantages over existing diffusion policy algorithms, notably in low-level control tasks. When benchmarked against the standard benchmark of offline reinforcement learning D4RL, DreamFuser matches or even outperforms contemporary methods.

## 1 Introduction

In the domain of Offline Reinforcement Learning, we introduce the DreamFuser algorithm aimed at seamlessly integrating Q-learning with trajectory-based diffusion policy. This algorithm leverages an innovative structure designed to resolve the inherent incompatibility between these promising features at a more refined level of granularity. We would introduce *DreamFuser*, an approach to trajectory-based value optimization using the diffusion model and *Generalized Noisy Action Markov Decision Process (GNMDP)*, a structure indicating the motivation of our algorithm.

DreamFuser harmoniously combines the advantages of two pivotal techniques: diffusion-based sequence modeling and efficient Q function learning, which encompass both state information and noisy actions. In light of the computational complexities arising from action sampling within the diffusion policy during training, we've devised the DreamFuser based on the framework of the GNMDP. Within this framework, we treat the hidden sampling process of the diffusion policy as an integral component of the Markov Decision Process (MDP) transition. Specifically, we could view the denoising step as a transition, where the *state* of this transition comprises the current state and the noisy action to be denoised, while the *action* of this transition is the predicted noise or equivalently the predicted denoised action. Thus, each step of the new GNMDP only necessitates a single denoising step or inherits the transition of the original MDP. Meanwhile, the properties of the Markov process are preserved. This enables us to define value and Q functions over observations and noisy action sequences so that the rewards could be maximized through optimizing the denoising network.

The benefit of GNMDP lies in its capacity to apply the actor-critic algorithm over a finer granularity without theoretical approximation. Instead of learning the Q-function solely over state and action, we learn it over state and noisy action. By focusing on optimization within a denoising step rather than across the entire sampling process, we can utilize multi-step temporal learning over GNMDP to balance the Q-function's variance and training speed. Drawing inspiration from TD3+BC (Fujimoto & Gu, 2021), we adapt this idea to GNMDP, resulting in DreamFuser.

The DreamFuser offers several advantages. Firstly, it preserves the structure of trajectory-based policies, facilitating more effective learning from sub-optimal and multimodal demonstrations. Si-

multaneously, integrating the Q-function empowers it to optimize action sequences, thereby improving rewards. Notably, applying the Q-function to the noisy action sequence significantly reduces computational demands, especially in resource-constrained scenarios. Unlike previous methods using pretrained value functions or approximating denoised actions, our method's trained Q-function aligns with the target policy and remains unbiased since it operates over noisy actions.

Moreover, DreamFuser has the potential to incorporate a learned dynamic model, further enhancing the learning process related to action sequences. As demonstrated in model-based methods like MOPO (Yu et al., 2020), MOReL (Kidambi et al., 2021), and COMBO (Yu et al., 2022), leveraging a dynamic model can exploit information from the joint distribution of states and actions. Thus, integrating a pretrained dynamics model injects transition features into our approach additionally.

We conducted a meticulous study to delineate the contributions of each component and empirically showcased the advantages of our approach on standard benchmarks. In summary, our method's contributions can be highlighted as follows:

- Seamlessly integrating trajectory-based diffusion policies with efficient Q learning, enabling the optimization of diffusion policies in offline RL scenarios.
- Resolving the incompatibility between these promising features through a finer granular perspective based on the versatile GNMDP structure, which is general and decoupled from any specific RL algorithm.
- Proposing the DreamFuser algorithm based on GNMDP, backed by comprehensive empirical evaluations that validate its effectiveness and demonstrate advantages using standard benchmark environments.

## 2 RELATED WORK

**Offline Reinforcement Learning** Offline Reinforcement Learning(RL) seeks to learn the optimal policy from a fixed and static dataset without interaction with the environment, which *offline* accounts for (Kumar et al., 2020) (Fujimoto et al., 2019). The datasets for offline RL usually comprise observations from the environment, actions of the agent, and the reward for the transition from the environment. The challenges in offline RL lie in the gap between the transition distribution of the offline dataset and the real distribution of interaction with the environment (Kostrikov et al., 2021). The gap is inevitable since the collected demonstrations are impossible to cover the whole state-action space in most cases. This gap poses extrapolation errors induced by actions from the learned policy if they are outside the behavior policy, i.e., the action distribution in the dataset. The bias of actions would be propagated through transitions and cause the agent to face totally unknown states.

Previously, the field has proposed different methods to restrict the learned action distribution close to the behavior policy. BEAR (Kumar et al., 2019) and BRAC (Wu et al., 2019) use regularization terms, like maximum mean discrepancy (MMD) or KL divergence between the learned policy and behavior policy, to restrict the learned policy distribution. Among them, TD3+BC (Fujimoto & Gu, 2021) uses the behavior-cloning loss term for policy regularization. Our approach builds upon the basic TD3+BC framework with a BC regularization term, taking advantage of its competent performance and compatibility with our backbone.

**Diffusion model in offline RL** As proposed by Chi et al. (2023), Chen et al. (2023), and Pearce et al. (2023), a policy learner with limited expressive capability could struggle to capture the distribution from the demonstrations, especially the demonstrations from the real world, which reveal multi-modality and complex distribution (Chi et al., 2023). Using a conventional uni-modal approximator would capture the average of behavior distributions and degrade the information learned from the datasets. For an informal instance (Chen et al., 2023), if an agent is faced by a wall, it can go either left or right to bypass it. However, if the leftward and rightward actions are comparable in the demonstration, it will learn to stay static according to the maximum likelihood estimation rule.

The diffusion model has been proposed as an expressive policy in recent works (Janner et al., 2022; Chi et al., 2023; Chen et al., 2023; Wang et al., 2022). The recent triumphs of diffusion models in generative modeling, encompassing diverse data modalities such as images (Rombach et al., 2022), audio (Kong et al., 2021), and video (Saharia et al., 2022), have ignited the advancement of diffusion policies that harness diffusion models to sample either single actions (Pearce et al., 2023) or

sequences of actions (Janner et al., 2022). The actions are denoised through the reverse diffusion process and conditional on the states. Given its nature as a generative model, the diffusion model possesses the capability to effectively characterize the multi-modal distributions (Shafiullah et al., 2022; Chi et al., 2023) of actions within the diverse policy learning datasets.

**Diffusion policy for sequence modeling** Sequence modeling has shown its promising characteristics in reinforcement learning. Sequence modeling over observation sequences and action sequences has the capacity to capture temporal correlations in long-horizon behavior policies. Chen et al. (2021) leverages the powerful transformer architectures as a sequence modeling backbone and has verified the power of trajectory-based learning. Chi et al. (2023) points out the promising role of sequence modeling in real-world deployment.

The intuitive combination of sequence modeling and diffusion model gave birth to the diffusers (Janner et al., 2022). The benefits come from the interplay between the sequence modeling and diffusion model. On the one hand, the diffusion model shows superior ability in modeling the sequence distribution (Vaswani et al., 2017; Janner et al., 2022; Ajay et al., 2022) to harness extensive data resources. On the other hand, trajectory prediction could amortize the notorious computation cost by sampling the diffusion model (Ho et al., 2020).

**Diffusion policy with Q(value)-function** In offline reinforcement learning, the challenge also lies in improving policies based on sub-optimal trajectories. Merely replicating the behavior distribution does not guarantee the maximization of rewards in such scenarios. Therefore, there is a necessity to enhance the diffusion policy in conjunction with behavior cloning objectives. In previous works like Janner et al. (2022) or Chi et al. (2023), a pretrained value function is learned to guide the sampling of the diffusion policy. However, the Q-function of the behavior policy is biased as a role of the learned policy, and the gradient of the Q-function could not align with the improved direction of the noisy actions in the intermediate denoising steps.

Alternatively, Diffusion-QL (Wang et al., 2022) employs an actor-critic framework to learn the Q function and policy alternatively. However, in the actor-critic training, both the policy evaluation and policy improvement phases would require sampling actions given the current observations. The back-propagation of action gradients throughout the reverse diffusion process is deemed impractical in training. Besides, Diffusion-QL (Wang et al., 2022) focuses on reinforcement learning over one time step. The integration of sequence modeling or trajectory-based learning would exaggerate the dilemma. This underscores the pressing need for a novel algorithm capable of optimizing action sequences while retaining the advantages of a diffusion model. Following works like EQP (Kang et al., 2023) alleviate the challenges through the approximation of denoised action, at a cost of precision.

## 3 PRELIMINARY

### 3.1 MDP AND OFFLINE RL

The Markov Decision Process (MDP) is denoted as $\mathcal{M} = \{\mathcal{S}, \mathcal{A}, P, R\}$. $\mathcal{S}$ and $\mathcal{A}$ represent the state and action spaces, respectively. We employ the subscript $t$ to signify the MDP step. The transition probability from state $s_t \in \mathcal{S}$ executing action $a \in \mathcal{A}$ and reach state $s_{t+1} \in \mathcal{S}$ is captured by $P(s_{t+1}|s_t, a)$, while $r_t = R(s_t, a, s_{t+1})$ quantifies the reward associated with the transition.

The goal in reinforcement learning is to find an optimal policy $\pi^*$, which is a conditional distribution of action, that maximizes the expected cumulative reward under the discount factor $\gamma \in [0, 1)$, i.e.

$$\pi^* = \arg\max_{\pi} \mathbb{E}[\sum_{t=0}^{\infty} \gamma^t r_t] \tag{1}$$

We can train a parameterized Q function, which approximates the expected cumulative rewards of a state-action pair, for a policy $\pi(a|s)$ by minimizing the bellman residual of sampled transitions, i.e., for transition $(s_t, a_t, s_{t+1}, r_t)$ we have

$$\mathcal{L} = \left( \left( r_t + \gamma E_{a_{t+1} \sim \pi} Q(s_{t+1}, a_{t+1}) \right) - Q(s_t, a_t) \right)^2 \tag{2}$$

In the offline setting, the environment is not accessible. Instead of online interactions with the environments, a static dataset $D = (s, a, r, s')$ collected by optimal or sub-optimal policies is used for policy learning.

## 3.2 DIFFUSION MODEL

Diffusion-based generative models (Sohl-Dickstein et al., 2015; Song et al., 2022; Ho et al., 2020) construct a sequence of diffusion steps that incrementally introduce noise while learning to regenerate the original samples from their noisier counterparts. Given an original sample $x \sim q(x)$ drawn from the true data distribution, a forward process employs a noise schedule $\beta = (\beta_1, \ldots, \beta_K)$, applying Gaussian noise $\epsilon_k$ iteratively for $K$ steps, thereby yielding a sequence of noisy samples $x_1, \ldots, x_K$. This process is formalized as (Wang et al., 2022):

$$q(\boldsymbol{x}^k|\boldsymbol{x}^{k-1}) = \mathcal{N}(\boldsymbol{x}^k; \sqrt{1-\beta_k}\boldsymbol{x}^{k-1}, \beta_k \boldsymbol{I}), \quad q(\boldsymbol{x}^{1:K}|\boldsymbol{x}^0) = \prod_{k=1}^{K} q(\boldsymbol{x}^k|\boldsymbol{x}^{k-1}) \tag{3}$$

Here, the superscript $k$ denotes the diffusion timestep, serving to differentiate it from the MDP timestep. Furthermore, as $K$ approaches infinity and $\beta_K$ approaches 1, the distribution of the noisy samples $\boldsymbol{x}^K$ becomes indistinguishable from $\mathcal{N}(\boldsymbol{0}, \boldsymbol{I})$.

Conversely, the reverse process initiates from pure Gaussian noise, $x_K \sim \mathcal{N}(\boldsymbol{0}, \boldsymbol{I})$, and is articulated as follows:

$$p_\theta(x_{k-1} \mid x_k) = \mathcal{N}(x_{k-1}; \mu_\theta(x_k, k), \Sigma_k) \tag{4}$$

In practice, we use non-parameterized standard variance $\Sigma_k = \tilde{\beta}_t I$ and $\tilde{\beta}_t = \frac{1-\bar{\alpha}_{t-1}}{1-\bar{\alpha}_t} \cdot \beta_t$. In alignment with DDPMs (Ho et al., 2020), we uses a reparameterized denoising function $\epsilon_\theta(x_k, k)$ such that $\mu_\theta(x^k, k) = \frac{1}{\sqrt{\alpha_k}}\left(x^k - \frac{1-\alpha_k}{\sqrt{1-\bar{\alpha}_k}}\epsilon_\theta(x^k, k)\right)$. The learning objective is defined as:

$$L_\mu = \mathbb{E}_{k \sim [1,K], \mathbf{x}_0 \sim p(x), \epsilon_k \sim \mathcal{N}(\boldsymbol{0}, \boldsymbol{I})}\left[(\boldsymbol{\epsilon}_k - \boldsymbol{\epsilon}_\theta(\sqrt{\bar{\alpha}_k}\mathbf{x}_0 + \sqrt{1-\bar{\alpha}_k}\boldsymbol{\epsilon}_k, k))^2\right] \tag{5}$$

Here, $\alpha_k = 1 - \beta_k$ and $\bar{\alpha}_k = \prod_{i=1}^{k} \alpha_i$.

## 4 METHOD

Our method, DreamFuser, trains trajectory-based diffusion policies to sample a sequence of actions based on past observations. In Section 4.1, we describe the formulation of our policy and then explain the difficulty of applying it in the offline RL. Following this, we then present a novel formulation in Section 4.2 that combines the diffusion process and the original MDP, which facilitates efficient Q learning and policy improvement in Section 4.3.

### 4.1 TRAINING DIFFUSION POLICIES IN OFFLINE RL

**Formulation of the diffusion policy** Given an MDP $\mathcal{M} = \{\mathcal{S}, \mathcal{A}, P, R\}$, we have a policy $\pi(\mathbf{A}_t|\mathbf{O}_t)$ determines a distribution over sequence of future actions $\mathbf{A}_t = \{a_t, a_{t+1}, \ldots, a_{t+l-1}\} \in \mathcal{A}^l$ conditioning on the past observations $\mathbf{O}_t = \{o_{t-h+1}, \ldots, o_t\} \in \mathcal{S}^h$ of length $h$ at time step $t$. For clarity, we focus predominantly on the trajectory form as a more general case with $\mathbf{O}_t$ and $\mathbf{A}_t$ in the following discussion. The action sequence $\mathbf{A}_t$ will be executed sequentially to obtain $\{o_{t+1}, o_{t+2}, \ldots, o_{t+l}\}$. One can perceive policy $\pi$ as integral to a new MDP, expressed as $\mathcal{M}' = \mathcal{S}', \mathcal{A}', P', R'$, where $\mathcal{S}' = \mathcal{S}^h$, $\mathcal{A}' = \mathcal{A}^l$. Further, we have $P'(\mathbf{O}_t, \mathbf{A}_t) = P'(\{o_{t-h+1}, \ldots, o_t\}, \{a_t, a_{t+1}, \ldots, a_{t+l-1}\}) = \{o_{t+l-h}, \ldots, o_{t+l}\} = \mathbf{O}_{t+l}$, $R'(\mathbf{O}_t, \mathbf{A}_t) = \sum_{i=0}^{l-1} r_{t+i}$ or $\sum_{i=0}^{l-1} \gamma^i r_{t+i}\}$ given discount factor $\gamma$. $\mathcal{M}'$ can be viewed as trajectory-based MDP.

As shown in Figure1, the diffusion model representation characterizes the distribution of one-step denoised action sequence $\mathbf{A}_t^{k-1}$ given observation sequence $\mathbf{O}_t$ and noisy action sequence $\mathbf{A}_t^k$:

$$\boldsymbol{\mu}_{\theta,\mathbf{O}_t}(\mathbf{A}_t^k, k) = \frac{1}{\sqrt{\alpha_k}}\left(\mathbf{A}_t^k - \frac{1-\alpha_k}{\sqrt{1-\bar{\alpha}_k}}\epsilon_{\theta,\mathbf{O}_t}(\mathbf{A}_t^k, k)\right),$$
$$\mathbf{A}_t^{k-1} \sim \mathcal{N}(\mathbf{A}_t^{k-1}; \boldsymbol{\mu}_{\theta,\mathbf{O}_t}(\mathbf{A}_t^k, k), \boldsymbol{\Sigma}_k), \tag{6}$$

where $\theta$ denotes the parameter of the diffusion model, $\boldsymbol{\mu}_{\theta,\mathbf{O}_t}$ is the mean of the denoised action distribution based on the noise predicted by $\epsilon_{\theta,\mathbf{O}_t}$, and $\boldsymbol{\Sigma}_k$ denotes the covariance matrix. The policy $\pi(\mathbf{A}_t \mid \mathbf{O}_t)$ involves sequential denoising steps and ultimately yields $\mathbf{A}_t = \mathbf{A}_t^0$ derived from $\mathbf{A}_t^K \sim \mathcal{N}(0, \mathbf{I})$ given the current state $\mathbf{O}_t$. The variable $\alpha_k$ and $\bar{\alpha}_k$ controls the variance as described Section 3.2.

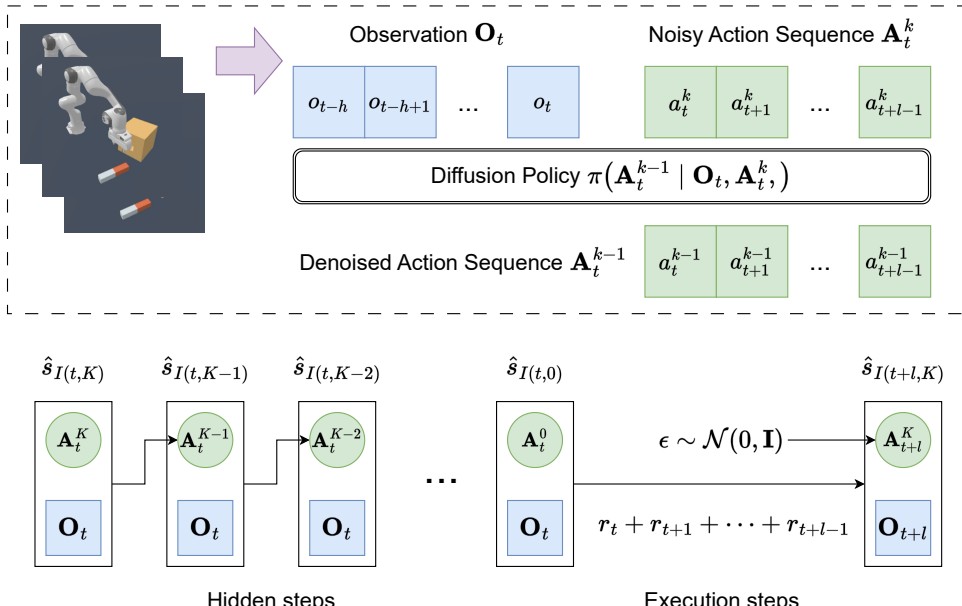

Figure 1: Illustrative of the diffusion policy and our GNMDP. Above: the diffusion policy that refines the noisy action sequences based on a sequence of past observations. Below: the Generalized Noisy Action MDP which incorporates the sampling process as a part of the MDP as *hidden steps* followed by an *execution step*.

**Problems in optimizing diffusion policies** In supervised learning, training the diffusion policy is accomplished by maximizing an approximation of the log-likelihood of the ground-truth action sequences, conditioned on past observation pairs. This process is achieved by minimizing the following objective function:

$$E_{k\sim[1,K],\epsilon_t\sim\mathcal{N}(\mathbf{0},\mathbf{I}),\mathbf{O}_t,\mathbf{A}_t}[\omega_k\|\epsilon_{\theta,\mathbf{O}_t}(\mathbf{A}_t^k,k)-\epsilon_t\|^2],$$

where $\omega_k$ refers to $\frac{(1-\alpha_k)^2}{2\alpha_k(1-\bar{\alpha}_k)\|\mathbf{\Sigma_k}\|_2^2}$, and can be ignored in practice implementation(Ho et al., 2020). These formulations enable us to prevent the expensive back-propagation through the full diffusion process.

However, optimizing policies with reinforcement learning requires a different learning scheme. The policy gradient theorem needs to evaluate $E_{\mathbf{A}_t\sim\pi(\mathbf{A}_t|\mathbf{O}_t)}[Q(\mathbf{O}_t,\mathbf{A}_t)\nabla\log\pi(\mathbf{A}_t|\mathbf{O}_t)]$ where the Q function $Q(\mathbf{O}_t,\mathbf{A}_t)$ approximates the expected rewards of generating action sequence $\mathbf{A}_t$ under observation sequence $\mathbf{O}_t$. This objective requires the actions sampling from the current diffusion model, making the sampling inefficient and requiring significant memory in the back-propagation.

## 4.2 GENERALIZED NOISY ACTION MDP

To avoid the back-propagation along the diffusion process, we introduce the concept of the **Generalized Noisy Action MDP**, abbreviated as **GNMDP**. The GNMDP views the diffusion policy's diffusion process as part of the original MDP structure. As illustrated in Figure 1, we insert $K$ new steps before the original transition as hidden transitions, mirroring the sampling process that progressively refines noisy action sequences into actions executed in the initial MDP. Although the new MDP requires additional steps to go through the same trajectory, it ensures that in each transition of the new MDP the policy needs to either denoise a noisy action sequence or execute an action sequence. This approach avoids fully denoising the noisy actions through the reverse diffusion process. We call the policy over GNMDP the *"denoise"* policy $\hat{\pi}$, which can be easily trained by any off-policy RL algorithms after collecting a batch of transitions.

Formally, we define $I(t,k)=(K+1)t+(K-k)$ to indicate the time index in the new MDP, where $K$ is the number of diffusion steps and $0\leq k\leq K$. A new state consists of the original state $s_t$

with observation $\mathbf{O}_t$ and a noisy action $\mathbf{A}_t^k$, i.e., $\hat{\boldsymbol{s}}_{I(t,k)} \triangleq (\mathbf{O}_t, \mathbf{A}_t^k)$. As shown in Figure 1, the step with timestamp $t$ in the original MDP is extended to $K+1$ steps with timestamps $I(t, K), I(t, K-1), \ldots, I(t, 0)$ respectively in GNMDP.

In GNMDP, for steps with $k > 0$, the denoising policy over GNMDP $\hat{\pi}$ will denoise $\mathbf{A}_t^k$ to $\mathbf{A}_t^{k-1}$ based on the observation $\mathbf{O}_t$, that is, generate action sequence $\hat{\boldsymbol{a}}_{I(t,k)} = \mathbf{A}_t^{k-1}$ from $\hat{\boldsymbol{s}}_{I(t,k)} = (\mathbf{O}_t, \mathbf{A}_t^k)$. We call these steps the denoise steps or *hidden steps*. Upon reaching $k = 0$, we will have $\mathbf{A}_t^0$, serving as the counterpart of $\mathbf{A}_t$ in the MDP context. The action sequence $\mathbf{A}_t^0$ is executed similarly to how $\mathbf{A}_t$ is executed in the original MDP, resulting in a transition to the next observation sequence $\mathbf{O}_{t+l}$. We refer to this as an *execution step*. During this step, a new noisy action sequence denoted as $\epsilon$, is sampled from $\mathcal{N}(0, \mathbf{I})$. The new noisy sequence is then combined with $\mathbf{O}_{t+l}$ to form the new augmented state $\hat{\boldsymbol{s}}_{I(t+l,K)} = (\mathbf{O}_{t+l}, \epsilon)$ in the execution step as shown in Figure 1. The associated reward for this transition is $R_t = R(\mathbf{O}_t, \mathbf{A}_t)$ in normal MDP, as mentioned in Section 4.1. More details over GNMDP are left in D.

## 4.3 Offline Reinforcement Learning in GNMDP

**Q function learning**  We are now ready to apply the actor-critic framework to train our diffusion policy. Q function is learnt over GNMDP to guide the optimization of policy at *hidden steps* and *execution steps*. Recalling the Q function loss defined in Equation 2, for GNMDP and $k > 0$, we have transition of form $(\hat{\boldsymbol{s}}, \mathbf{A}_t^{k-1}, \hat{\boldsymbol{s}}')$ where $\hat{\boldsymbol{s}} = \hat{\boldsymbol{s}}_{I(t,k)} = (\mathbf{O}_t, \mathbf{A}_t^k)$ and $\hat{\boldsymbol{s}}' = \hat{\boldsymbol{s}}_{I(t,k-1)} = (\mathbf{O}_t, \mathbf{A}_t^{k-1})$. Thus the loss take the form of

$$\mathcal{L}_{consistency} = \left( \gamma_2 \mathbb{E}_{\mathbf{A}_t^{k-1} \sim \hat{\pi}_\theta(\cdot|\hat{\boldsymbol{s}})} \hat{Q}_\phi(\hat{\boldsymbol{s}}', \mathbf{A}_t^{k-1}) - \hat{Q}_\phi(\hat{\boldsymbol{s}}, \mathbf{A}_t^k) \right)^2, \tag{7}$$

where $\gamma_2 \approx 1$ is the discount factor of *hidden steps*.

For $k = 0$, the bellman residual in GNMDP is of the same meaning as that in normal MDP. Let $\hat{\boldsymbol{s}} = \hat{\boldsymbol{s}}_{I(t,0)} = (\mathbf{O}_t, \mathbf{A}_t)$ and $\hat{\boldsymbol{s}}' = \hat{\boldsymbol{s}}_{I(t+l,K)} = (\mathbf{O}_{t+l}, \epsilon), \epsilon \sim \mathcal{N}(0, \mathbf{I})$, we have the **MSBE loss**(Mean Square Bellman Error loss):

$$\mathcal{L}_{MSBE} = \left( \left( \mathcal{R}_t + \gamma \hat{Q}_\phi(\hat{\boldsymbol{s}}', \epsilon) \right) - \hat{Q}_\phi(\hat{\boldsymbol{s}}, \mathbf{A}_t) \right)^2. \tag{8}$$

**Policy improvement**  Like TD3+BC, we combine a **Q loss** and a **BC loss** to train the policy. The **Q loss** pertains to the optimization term, and it can be expressed as:

$$\mathcal{L}_Q = -\hat{Q}_\phi(\hat{\boldsymbol{s}}_{I(t,k)}, \hat{\boldsymbol{a}}_{I(t,k)}) = -\hat{Q}_\phi((\mathbf{O}_t, \mathbf{A}_t^k), \mathbf{A}_t^{k-1}). \tag{9}$$

The **BC loss** serves as the regression term for preventing out-of-distribution actions, which could be replaced by the loss of the diffusion model:

$$\mathcal{L}_{BC} = \left( \epsilon_{t,k} - \epsilon_{\theta, \mathbf{O}_t}(\mathbf{A}_t^k, k) \right)^2 \tag{10}$$

$\epsilon_{t,k}$ refers to the sampled diffusion noise corresponding to the $I(t, k)$ step of GNMDP. That is the noise sampled in the forward diffusion process of $\mathbf{A}_t$ to $k$-th step. The coefficients of the Q loss and BC loss can be fine-tuned to balance the optimization goal and constraints on the action distribution. Our method, DreamFuser, optimizes a diffusion policy in GNMDP. The training process involves sampling trajectories from the dataset, training the Q networks, and optimizing the "denoise" policy. The inference process includes sampling the action sequence through the diffusion model of DreamFuser and applying it sequentially to obtain the next observations. Detailed steps of the proposed algorithm are presented in Appendix Algorithm 1.

**Pretrained dynamic model**  Optionally, we incorporate a dynamics model to enhance the policy's prediction capabilities. Complementary to our main algorithm, the dynamics model is equipped with a GRU backbone. It has been pre-trained on the behavior dataset to predict future state transitions, represented as $O_t'$. The inputs of the dynamics model include the noisy actions $A_t^k$, the previous state $O_t$, and the diffusion timestep $k$. The superscript $k$ on $O_t'^k$ indicates the diffusion timestep associated with its corresponding input, $A_t^k$. The integration of the pre-trained dynamics model

allows us to enhance the policy's input with the predicted future state $O_t^{\prime k}$, as the dynamics model captures the transition features of the environment. In other words, the diffusion policy inputs $O_t$, $k$, $O_t^{\prime k}$, $A_t^k$, and outputs $A_t^{k-1}$ in the case with a pretrained dynamics model. For a detailed exposition of the dynamics model, including its integration and functionality, please refer to Appendix Section B.3.

## 5 EXPERIMENTS

This section showcases our method's ability to efficiently learn the Q function to optimize the diffusion policy within the context of GNMDP. Our approach harnesses the advantages of trajectory-based diffusion models and the learned Q function, thereby preserving the effectiveness of sequence modeling during behavior cloning while enhancing diffusion policies in offline RL. We conduct ablation studies to underscore the contributions of our approach's individual components.

### 5.1 EVALUATION BENCHMARK

**D4RL** Our experimental assessment of the proposed algorithm is conducted on the D4RL (Fu et al., 2021) benchmark, specifically focusing on the Gym locomotion and AntMaze environments. The dataset for each environment comprises 1e6 timesteps. The locomotion datasets encompass a notable number of near-optimal trajectories with dense rewards. The AntMaze datasets contain very few near-optimal trajectories and are primarily characterized by sparse rewards that are hard to optimize. Successful navigation toward the maze's intended goal requires the agent to effectively integrate various sub-optimal trajectories.

**Object manipulation** The evaluation also emphasizes low-level object manipulation tasks, as we observe saturation in the D4RL benchmarks for imitation learning. This focus is essential to more effectively highlight the advantages of trajectory learning and diffusion model capabilities. We take two environments from ManiSkill2 (Gu et al., 2023) as our testbed. ManiSkill2 offers diverse object manipulation tasks executed in environments with authentic physical simulation, encompassing dynamic grasping motions. The task *Peg Insertion Side* requires the agent to insert a cuboid-shaped peg sideways into a hole of varied geometries and sizes, embodying a precise clearance task. Another task *Stack Cube*, a 6-DoF Pick-and-place task, requires the agent to pick up and lift a cube, and position it atop another. *Block Pushing*, derived from BET (Shafiullah et al., 2022), evaluates the policy's proficiency in handling multimodal action distributions through the objective of pushing two blocks into two squares in no specific order, underlining the task's multimodal nature. The above datasets all contain 1k trajectories with maximum length not exceeding 50.

### 5.2 BASELINE COMPARISON

We incorporate comparisons with offline RL baselines within the realm of regularization-based methodologies. This includes Behavior Cloning (BC), TD3 augmented with BC (Fujimoto & Gu, 2021), and Implicit Q-Learning (IQL) (Kostrikov et al., 2021). Furthermore, we also consider Q-value constraint methodologies, specifically Conservative Q-Learning (CQL) (Kumar et al., 2020). We also compare trajectory modeling algorithms like BeT (Shafiullah et al., 2022) and diffusion-based sequence modeling methods such as Diffuser (Janner et al., 2022), Decision Diffusers (DD) (Ajay et al., 2022), and Diffusion Policy (Chi et al., 2023) for comparison. The performance metrics of these baseline algorithms are based on either the best results recorded in their respective publications(for D4RL) or results reproduced by us(for manipulation tasks).

**Results for D4RL** The results 1 highlight that while baselines yield satisfactory performance on the Gym MuJoCo tasks, our proposed algorithm consistently enhances these performances, as evidenced by the averaged normalized scores. Particularly noteworthy are the tasks classified as "medium" where our method outperforms all the baselines listed. These "medium" datasets inherently contain trajectories generated by agents exhibiting exploratory behaviors and deploying non-optimized policies. The mere imitation of the data distribution will lead to sub-optimal policy. As elucidated in Section 4, our method, DreamFuser, navigates through the policy improvement phase, generating better actions based on the non-optimal sequences through Q-function guided optimization process. When considering the AntMaze tasks, it is needed to apply robust and stable Q-learning to adeptly connect sub-optimal demonstrations, facilitating the agent's progression to-

Table 1: Averaged normalized scores on MuJoCo locomotion and Ant Maze tasks. Algorithms demonstrating the top two performances are highlighted in the table. The locomotion tasks are evaluated with 10 seeds and the Ant Maze tasks are measured with 3 seeds. For each seed, 20 episodes are evaluated

| Gym Task | BC | TD3+BC | Diffuser | IQL | DD | DP | Ours |
|---|---|---|---|---|---|---|---|
| halfcheetah-medium-v2 | 42.6 | 42.6 | 44.2 | 47.4 | **49.1** | 41.7 | **52.8** $\pm$ 0.0 |
| hopper-medium-v2 | 52.9 | 67.6 | 58.5 | 66.3 | **79.3** | 51.1 | **93.8** $\pm$ 3.5 |
| walker2d-medium-v2 | 75.3 | 74.0 | 79.7 | 78.3 | **82.5** | 77.2 | **85.9** $\pm$ 0.3 |
| halfcheetah-medium-expert-v2 | 55.2 | 86.8 | 79.8 | 86.7 | **90.6** | 43.5 | **94.8** $\pm$ 0.0 |
| hopper-medium-expert-v2 | 52.5 | 107.6 | 107.2 | 91.5 | **111.8** | 53.6 | 106.3 $\pm$ 3.5 |
| walker2d-medium-expert-v2 | 107.5 | 108.1 | 108.4 | **109.6** | 108.8 | 77.2 | **109.5** $\pm$ 0.0 |
| **Average Score** | 64.3 | 81.11 | 79.6 | 80.0 | **87.0** | 57.3 | **90.5** |
| **AntMaze Tasks** | BC | TD3+BC | CQL | IQL | - | DP | Ours |
| antmaze-umaze-v0 | 54.6 | 78.6 | 74.0 | **87.5** | - | 64.0 | **92.1** |
| antmaze-umaze-diverse-v0 | 45.6 | 71.4 | **84.0** | 62.2 | - | 60.0 | **82.7** |
| antmaze-medium-diverse-v0 | 0.0 | 3.0 | 53.7 | **70.0** | - | 2.0 | **70.6** |
| antmaze-large-diverse-v0 | 0.0 | 0.0 | 14.9 | **47.5** | - | 0.0 | **46.0** |
| **Average Score** | 25.0 | 38.3 | 56.7 | **66.4** | - | 31.5 | **72.9** |

Table 2: Averaged success rate (in %) on ManiSkill2 and multimodal Block Push tasks.

| Object Manipulation Tasks | BeT | DD | Diffusion Policy | Ours |
|---|---|---|---|---|
| PegInsertionSide-v0 | 0.8 | 13.0 | 50.3 | **53.5** |
| StackCube-v0 | 0.6 | 5.6 | 83.6 | **89.9** |
| BlockPush | 71 | 75 | **94** | 92.3 |
| **Average Score** | 24.1 | 31.2 | **76.0** | **78.6** |

wards the ultimate target. Our proposed trajectory learning optimization technique for Q values proves effective, notably excelling in tasks labeled as "umaze". In summary, the DreamFuser shows the advantages in optimizing diffusion policy, verified by the results of D4RL benchmark.

**Results for object manipulation task** Our methodologies distinctly surpass all benchmarked trajectory baselines 2 on ManiSkill2 tasks, irrespective of whether they are diffusion-based or not. When considering the Block Push task, DreamFuser achieves comparative results w.r.t state-of-the-art diffusion sequence-based method Diffusion Policy. The challenges posed by the necessity for high precision, the complexity of the control action space, or the multimodal nature of the data distribution are effectively navigated by our approach.

### 5.3 ABLATION STUDY

**Effectiveness of Q learning** In our ablation study, we investigate the impact of Q optimization on the performance of our proposed algorithm. The results are presented in Table 3, showcasing the performances with and without applying Q optimization. From the displayed results, it is clear that incorporating Q optimization substantially enhances the algorithm's efficacy across the selected tasks. In scenarios without Q optimization, the algorithm exhibits a reduction in performance, underscoring the pivotal role that Q optimization plays in bolstering the model's capabilities.

**Length of the action sequence** Within both experimental environments, the visualized results presented in Figure 2c and 2b indicate that, under conditions of a fixed input horizon, approaches employing trajectories modeling (horizon equals 7) exhibit substantially higher performance com-

Table 3: Averaged normalized scores on MuJoCo locomotion and averaged success rate on ManiSkill2 tasks. Our method has a strong performance boost after including Q optimization.

| Maniskill Task | w/o Q | w Q |
|---|---|---|
| PegInsertionSide-v0 | 49.3 | 53.5 |
| StackCube-v0 | 86.4 | 89.9 |
| **Gym Task** | **w/o Q** | **w Q** |
| halfcheetah-medium-v2 | 42.8 | 51.0 |
| hopper-medium-v2 | 52.7 | 94.7 |
| walker2d-medium-v2 | 56.4 | 88.5 |
| halfcheetah-medium-expert-v2 | 66.4 | 92.9 |
| hopper-medium-expert-v2 | 60.6 | 102.3 |
| walker2d-medium-expert-v2 | 86.9 | 109.7 |

pared to those utilizing a singular time step (where the output horizon is set to 1). This empirical observation underscores the substantial advantages of forecasting longer sequences when engaged in low-level control tasks that deal with manipulation and movement at a finer, more granular level. We conjecture that modeling longer action sequences helps the network capture the temporal correlation of the behavior policy, enabling more accurate modeling of the behavior distributions. As a result, modeling action sequences generates trajectories closer to the demonstrations and achieves a higher success rate.

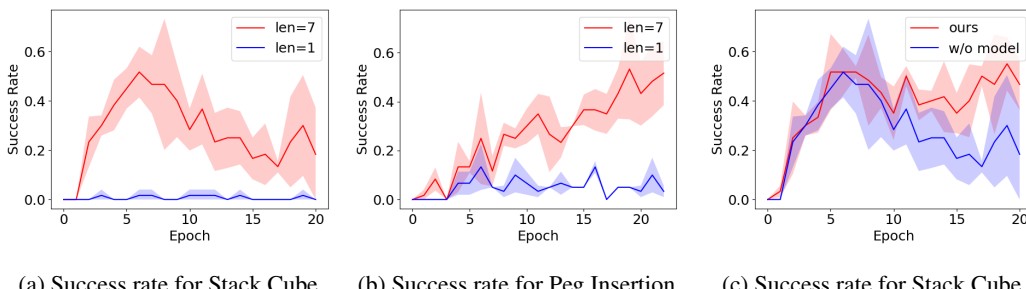

(a) Success rate for Stack Cube  (b) Success rate for Peg Insertion  (c) Success rate for Stack Cube

Figure 2: Performance curves for two tasks: Peg Insertion and Stack Cube. Each epoch corresponds to a 50k gradients update.

**Effectiveness of the learned model**   The pretrained dynamics model helps address the overfitting problem. The overfitting issue occurs in some low level control tasks and it is due to the limited capacity of low-level control dataset presumably. As depicted in Figure 2c, removing the learned model shows a discernible decline in performance beyond the 10th epoch, whereas the model-based policy continues to exhibit an ascending trajectory in terms of performance. This graphical representation elucidates that the adoption of a model-based approach can mitigates the overfitting issue as the learned model may help the network better generalize from the learned data patterns.

## 6   CONCLUSION AND FUTURE WORKS

This work introduced the DreamFuser algorithm, which seamlessly integrates the Diffusion Model into a Generalized Noisy Action MDP framework for offline reinforcement learning. This integration enhanced memory utilization and computational efficiency compared to existing methodologies while preserving the expressive capabilities of diffusion models on sequence modeling. As demonstrated by empirical results, DreamFuser attains improved performance on benchmark datasets.

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

## A CODE

---

**Algorithm 1** DreamFuser

---

**Require:** Given $D = \{s_t, a_t, r_t, d_t\}$, obtain trajectory dataset $D' = \{\mathbf{O}_t, \mathbf{A}_t, \mathcal{R}_t, \mathcal{D}_t\}$. Conditional Diffusion Model $\epsilon_\theta(\mathbf{O}_t, \mathbf{A}_t^k, k)$ with noising function $q\_sample$: $\mathbf{A}_t^k = q(\mathbf{A}_t, k, \epsilon)$, and denoising function $p\_sample$: $\mathbf{A}_t^{k-1} \approx p_\theta(\mathbf{A}_t^k, k, \mathbf{O}_t)$. $C_1$ and $C_2$ are constant coefficients. $r \in \{1, \ldots, K\}$ is the denoising rollout step number.

1: **while** not converged **do**
2:   $k \leftarrow [r, K], b = \{\mathbf{O}_t, \mathbf{A}_t, \mathbf{O}'_t, \mathcal{R}_t, \mathcal{D}_t\} \leftarrow D'$
3:   $\mathbf{A}_t^k = q(\mathbf{A}_t, k, \epsilon), \epsilon \leftarrow \mathcal{N}$
4:   **for** $i \in \{1, \ldots, r\}$ **do**
5:    $\bar{\mathbf{A}}_t^{k-i} = p_{\bar{\theta}}(\mathbf{A}_t^{k-i+1}, k-i+1, \mathbf{O}_t), \mathbf{A}_t^{k-i} = p_\theta(\mathbf{A}_t^{k-i+1}, k-i+1, \mathbf{O}_t)$
6:   **end for**
7:   $L_{MSBE} = \left\| \hat{Q}_\phi(\mathbf{A}_t, 0, \mathbf{O}_t) - (\mathcal{R}_t + (1-\mathcal{D}_t)\gamma \hat{Q}_{\bar{\phi}}(\epsilon', K, \mathbf{O}'_t)) \right\|^2, \epsilon' \leftarrow \mathcal{N}$
8:   $L_{consistency} = \left\| \hat{Q}_\phi(\mathbf{A}_t^k, k, \mathbf{O}_t) - \gamma_2 \hat{Q}_{\bar{\phi}}(\bar{\mathbf{A}}_t^{k-r}, k-r, \mathbf{O}_t) \right\|^2$
9:   $L_Q = -\hat{Q}_\phi(\mathbf{A}_t^{k-r}, k-r, \mathbf{O}_t)$
10:   $L_{BC} = \left\| \epsilon_\theta(\mathbf{O}_t, \mathbf{A}_t^k, k) - \epsilon \right\|^2$
11:   $L_{critic} = L_{consistency} + C_1 L_{MSBE}$
12:   $L_{actor} = L_Q + C_2 L_{BC}$
13:   $\phi \leftarrow \phi - \eta_\phi \nabla_\phi L_{critic}, \theta \leftarrow \theta - \eta_\theta \nabla_\theta L_{actor}$ if update needed
14:   **if** update needed **then**
15:    $\bar{\phi} = \tau\bar{\phi} + (1-\tau)\phi, \bar{\theta} = \tau'\bar{\theta} + (1-\tau')\theta$
16:   **end if**
17: **end while**

---

---

**Algorithm 2** DreamFuser Inference

---

**Require:** Given the current observations $\mathbf{O}_t$, the trained model $M$, $K_{inference}$ is the sampling steps of the DDIM.

1: $\epsilon \sim \mathcal{N}(0, \mathbf{I})$.
2: $\mathbf{A}_t$ is denoised conditional on $\mathbf{O}_t$ from $\epsilon$ in $K_{inference}$ steps by $M$ and following the DDIM denoising scheduler.

---

## B ADDITIONAL DETAILS

Here we provide additional details for our algorithm implementation.

### B.1 POLICY TRAINING

The training process consists of two phases: Q function learning(policy evaluation) and policy improvement.

We first sample trajectories from the provided dataset and add noise to get $\mathbf{A}_t^k$ as shown in the pseudocode 1. In the Q function learning phase, we minimize the **MSBE loss** 8 and **consistency loss** 7 to learn the Q function over GNMDP. The MSBE loss refers to the Mean Square Bellman update Errors between the steps in the original MDP while the consistency loss refers to the Bellman update loss between the denoising steps. We use the target network of actors to denoise noisy action in this phase to stabilize the training process. We can adjust the ratio between the coefficient of MSBE loss and consistency loss and determine a constant coefficient. The *double Q* method is applied for both *hidden steps* and *execution steps*, so as to make Q-learning more stable.

In the policy improvement phase, we minimize the **Q loss** and **BC loss** terms. The actor is used to denoise the noisy actions. The weight of BC loss can be tuned based on the environment. The Q loss accounts to optimize the diffusion policy towards the direction maximizing the objectives. The

BC loss is inspired by TD3+BC, which could would help control the distance between the target policy distribution and the behavior policy distribution. We keep the BC loss coefficient the same over most environments.

We introduce the multi-step value evaluation in our DreamFuser. For both actor and critic, we add target networks. Given rollout number $r$, we continuously denoise the noisy actions $\mathbf{A}_t^k$ for $r$ steps to get $\bar{\mathbf{A}}_t^{k-r}$ and $\mathbf{A}_t^{k-r}$, as shown in 1, which could be used to obtain a better value estimation in the computation of consistency loss and Q loss. The rollout number $r$ can be tuned to control the computation cost of denoising steps, enabling us to trade off between computation and performance. As we know, the multi-step temporal learning would reduce the variance of Q-function.

## B.2 INFERENCE OF THE DREAMFUSER

correct the grammar and keep the identifier in latex The inference of the DreamFuser is the same as the diffusion policy. At the beginning, a noise is sampled from the Normal distribution. The noise has the same shape as the action sequence and serve as the initial noisy action. The primary inputs to the diffusion model in the DreamFuser are a sequence of observations, denoted as $\mathbf{O}_t$, and a sequence of noisy actions, $\mathbf{A}_t^k$, as well as the diffusion step $k$. The $\mathbf{O}_t$ and $k$ serve as the condition of the diffusion model. Then the model would denoise and output $\mathbf{A}_t^{k-1}$.

To accelerate the policy's inference speed, we use Denoising Diffusion Implicit Models (DDIM)(Song et al., 2021) in sampling. DDIM is instrumental in accelerating the inference process by enabling the selection of a reduced number of diffusion steps, known as $K_{inference}$ instead of the original $K$. The training of DDIM is the same as the DDPM. Opting for a smaller $K_{inference}$ facilitates a swifter evaluation of the model, thereby enhancing efficiency without significantly compromising the quality of the generated actions. Thus, the above $\mathbf{A}_t^k$ and $\mathbf{A}_t^{k-1}$ of inference step are in the context where $k \in [1, K_{inference}]$ instead of $k \in [1, K]$. In practice, $K_{inferemce}$ is chosen to be 4 for faster evaluation, independently of the chosen value for $K$.

## B.3 MODEL-AUGMENTED ACTION GENERATION

The policy's capability is further enhanced by integrating a pre-trained dynamics model. That is, apart from the inputs mentioned in B.2, the output of the dynamics model could serve as an additional input. The pretrained dynamics model uses a GRU(Chung et al., 2014) backbone to predict future state transitions, denoted as $O_t'$. For the GRU model, we use the history observation sequence $O_t$ for the initial hidden and the noisy action sequence as input. The sequence length of future states $O_t'^k$ is chosen to be the same as the length of action sequence $\mathbf{A}_t$. Thus, the inputs of the diffusion model would be $\mathbf{O}_t, \mathbf{O}_t'^k, \mathbf{A}_t^k, k$ when equiped with the pretrained dynamics model. In the pretraining process of the dynamics model, we use the supervised learning in predicting the next observations given the current observations and actions sequence. The motivation of the additional dynamic model is to embed the knowledge about the transition of environments into the diffusion policy. We use the dynamics model in our low-level benchmarks evaluation.

## B.4 NETWORK ARCHITECTURE

For the dynamics model mentioned in 4.3 and B.3, the GRU is used as the backbone. The hidden dimension of the GRU is 256. The inputs $\mathbf{O}_t$ are fed into a conditional GRU, and the final hidden states are extracted as the features of the current observations $\mathbf{O}_t$. Then, the hidden states and time embedding of $k$ serve as the initial hidden states for the transition GRU, whose inputs are $\mathbf{A}_t^k$. Finally, the transition GRU will output $\mathbf{O}_t'^k$, which is further transferred to the diffusion model.

When it comes to the sequence-based denoising model, our approach adheres to the Transformer backbone as delineated in (Chi et al., 2023). In this setup, the embedding of observation $\mathbf{O}_t$ is combined with the embedding of the diffusion timestep $k$ by summation, and they are then fed into a multi-head cross-attention layer of each Transformer decoder block. Furthermore, the input for the Transformer decoder is formulated by concatenating the noisy action sequence $\mathbf{A}_t^k$ and the predicted future state $\mathbf{O}_t'^k$, which can be expressed as $[\mathbf{A}_t^k, \mathbf{O}_t'^k]$. The alternatives of the current backbones could be left as the future work.

## C  HYPERPARAMETERS

| Hyperparameter | Value |
|---|---|
| batch size | 256 |
| number of steps per epoch | 5000 |
| number of epochs | 2000 |
| learning rate | 3e-4 |
| max episode length | 1000 |
| evaluation episodes | 20 |

Table 4: Hyperparameter Configuration

When evaluating the policy, we tested over 3 seeds and 20 episodes for each. We use the average reward sum and success rate to quantify the performance. The average reward sum is used in the D4RL dataset while the success rate is referred to in the Maniskill2 dataset.

| Hyperparameter | Value |
|---|---|
| gamma | 0.99 |
| gamma2 | 1.0 |
| tau | 0.005 |
| policy update frequency | 2 |
| policy ema frequency | 5 |
| critic ema frequency | 1 |
| grad norm | 9.0 |
| state length | 3 |
| action length | 2 |
| length of rollout | 1 |
| MSBE coefficient | 1.0 |
| Q coefficient | 1.0 |
| BC coefficient | 1.0 |
| consistency coefficient | 1.0 |
| beta schedule | linear |
| predict epsilon | False |
| sampler type | ddim |
| diffusion steps | 8 |
| number of inference steps | 4 |

Table 5: Hyperparameter Configuration

The policy update frequency refers to the number of critical updates between policy updates. policy ema frequency and critic ema frequency refer to the frequency of updating the target network. The gradient norm is used in the gradient clip in the backward propagation. The state length, and action length are the number of states used as conditions and the number of actions predicted.

The default hyperparameters are listed above. For the D4RL dataset, over the Mujoco and Antmaze environments, we use the default hyperparameters for all these environments. The results of halfcheetah-medium-expert-v2 and walker2d-medium-expert-v2 are evaluated with state length and action length as 1.

For the Maniskill2 dataset, we use state length of 2, action length of 7, and diffusion steps as 100.

## D  DETAILS ABOUT GNMDP

### D.1  FORMULATION OF GNMDP

In order to present our approach, **DreamFuser**, we introduce the concept of the **Generalized Noisy Action MDP**, abbreviated as **GNMDP**. The GNMDP serves as the foundation for incorporating the diffusion policy's diffusion process into the underlying original MDP structure. A similar notion

has been put forth in Black et al. (2023), wherein the denoising procedure of the diffusion model is treated as an MDP, utilizing a reinforcement learning (RL) framework to optimize an objective function throughout the diffusion process. It's worth noting that, in our context, the primary emphasis lies within the domain of reinforcement learning. Hence, we integrate the diffusion process seamlessly into the RL framework of the original MDP, rather than constructing an MDP solely to encapsulate the diffusion process from scratch.

Let's consider the original MDP within the framework of reinforcement learning, denoted as $\mathcal{M} = \{\mathcal{S}, \mathcal{A}, P, R, \gamma\}$. Here, $\mathcal{S}$ and $\mathcal{A}$ represent the state and action spaces, respectively. The transition probability from state $\boldsymbol{s}$ to state $\boldsymbol{s}'$ following action $\boldsymbol{a}$ is captured by $P(\boldsymbol{s}'|\boldsymbol{s}, \boldsymbol{a})$, while $R(\boldsymbol{s}, \boldsymbol{a}, \boldsymbol{s}')$ quantifies the reward associated with the transition. Within this context, $\gamma \in [0, 1)$ serves as the discount factor. To distinguish from the diffusion timestep, we employ the subscript $t$ to signify the MDP timestep.

On a different note, in the context of the diffusion model, we adopt the superscript $k$ to represent the diffusion timestep for ease of reference. Given a real distribution $q(x)$, an initial sample $\boldsymbol{x}^0$ is drawn from $q(x)$. Progressively introducing Gaussian noise results in a sequence of noisy samples $\boldsymbol{x}^1, \ldots, \boldsymbol{x}^K$, with a noise schedule $\beta = (\beta_1, \ldots, \beta_K)$. This noise incorporation process is defined as follows:

$$q(\boldsymbol{x}^k|\boldsymbol{x}^{k-1}) = \mathcal{N}(\boldsymbol{x}^k; \sqrt{1 - \beta^k}\boldsymbol{x}^{k-1}, \beta^k \boldsymbol{I}) \tag{11}$$

$$q(\boldsymbol{x}^{1:K}|\boldsymbol{x}^0) = \prod_{k=1}^{K} q(\boldsymbol{x}^k|\boldsymbol{x}^{k-1}). \tag{12}$$

Moreover, as $K \to \infty$, $\beta_K \to 1$, the distribution of noisy samples $\boldsymbol{x}^K$ should become indistinguishable from $\mathcal{N}(\boldsymbol{0}, \boldsymbol{I})$.

We then introduce the subsequent MDP $\hat{\mathcal{M}} = \{\hat{\mathcal{S}}, \hat{\mathcal{A}}, \hat{P}, \hat{R}, \hat{\gamma}\}$, defined as follows:

$$\hat{\boldsymbol{s}}_{I(t,k)} \triangleq (\boldsymbol{s}_t, \boldsymbol{a}_t^k) \tag{13}$$

$$\hat{\boldsymbol{a}}_{I(t,k)} \triangleq \begin{cases} \boldsymbol{a}_t^{k-1}, & (k > 0), \\ \epsilon \sim \mathcal{N}(0, \mathbf{I}), & (k = 0), \end{cases} \tag{14}$$

$$\hat{P}(\hat{\boldsymbol{s}}_{I(t,k)}, \hat{\boldsymbol{a}}_{I(t,k)}) \triangleq \begin{cases} (\boldsymbol{s}_t, \boldsymbol{a}_t^{k-1}), & (k > 0), \\ (P(\boldsymbol{s}_t, \boldsymbol{a}_t^0), \epsilon), & (k = 0, \epsilon \sim \mathcal{N}(0, \boldsymbol{I})), \end{cases} \tag{15}$$

$$\hat{\boldsymbol{r}}_{I(t,k)} \triangleq \begin{cases} 0, & (k > 0), \\ \boldsymbol{r}_t, & (k = 0), \end{cases} \tag{16}$$

$$\hat{\gamma}_{I(t,k)} = \triangleq \begin{cases} \gamma_2, & (k > 0), \\ \gamma, & (k = 0), \end{cases} \tag{17}$$

Here, $I(t, k) = (K + 1)t + (K - k)$ indicates the time index in the new MDP, and $\gamma_2 = 1$. $k > 0$ step is referred as denoise or *hidden* step while $k = 0$ step is referred as *execution step*.

### D.2 POLICY AND VALUE FUNCTION OVER GNMDP

The policy $\pi(\boldsymbol{a} \mid \boldsymbol{s})$ pertains to the action distribution of $\boldsymbol{a}$ given the current state $\boldsymbol{s}$ in the original MDP. The corresponding objective function for this MDP is as follows:

$$\max_{\pi} \mathbb{E}\left[\sum_{t=0}^{\infty} \gamma^t r(s_t, a_t)\right]. \tag{18}$$

In addition, commonly used in reinforcement learning algorithms are the value function $V(\boldsymbol{s})$ and the action value (Q) function $Q(\boldsymbol{s}, \boldsymbol{a})$, defined respectively as:

$$V(\boldsymbol{s}) = \mathbb{E}_{\pi}\left[\sum_{t=0}^{\infty}\gamma^t r(\boldsymbol{s}_t, \boldsymbol{a}_t)|\boldsymbol{s}_0 = \boldsymbol{s}\right], \tag{19}$$

$$Q(\boldsymbol{s}, \boldsymbol{a}) = \mathbb{E}_{\pi}\left[\sum_{t=0}^{\infty}\gamma^t r(\boldsymbol{s}_t, \boldsymbol{a}_t)|\boldsymbol{s}_0 = \boldsymbol{s}, \boldsymbol{a}_0 = \boldsymbol{a}\right]. \tag{20}$$

Building upon the slightly revised definition of the GNMDP provided above, we can demonstrate that the policy $\hat{\pi}$, value function $\hat{V}$, and action value function $\hat{Q}$ are well-defined. Notably, the action at $k = 0$ has no effect. For clarity, we can alternatively define $\hat{\mathcal{M}} = \{\hat{\mathcal{S}}, \hat{\mathcal{A}}, \hat{P}, \hat{R}, \hat{\gamma}\}$ as follows:

$$\hat{\boldsymbol{s}}_{I(t,k)} \triangleq (\boldsymbol{s}_t, \boldsymbol{a}_t^k), \quad (1 \le k) \tag{21}$$

$$\hat{\boldsymbol{a}}_{I(t,k)} = \boldsymbol{a}_t^{k-1}, \quad (k > 0), \tag{22}$$

$$\hat{P}(\hat{\boldsymbol{s}}_{I(t,k)}, \hat{\boldsymbol{a}}_{I(t,k)}) = \hat{P}((\boldsymbol{s}_t, \boldsymbol{a}_t^k), \boldsymbol{a}_t^{k-1}) \triangleq \begin{cases} (\boldsymbol{s}_t, \boldsymbol{a}_t^{k-1}), & (k > 1), \\ (P(\boldsymbol{s}_t, \boldsymbol{a}_t^0), \epsilon), & (k = 1, \epsilon \sim \mathcal{N}(0, \boldsymbol{I})), \end{cases} \tag{23}$$

$$\hat{\boldsymbol{r}}_{I(t,k)} \triangleq \begin{cases} 0, & (k > 1), \\ \boldsymbol{r}_t, & (k = 1), \end{cases} \tag{24}$$

$$\hat{\gamma}_{I(t,k)} =\triangleq \begin{cases} \gamma_2, & (k > 1), \\ \gamma\gamma_2, & (k = 1), \end{cases} \tag{25}$$

Subsequently, $\hat{\pi}(\hat{\boldsymbol{a}} \mid \hat{\boldsymbol{s}}) = \hat{\pi}(\boldsymbol{a}_t^{k-1} \mid \boldsymbol{s}_t, \boldsymbol{a}_t^k)$ represents the distribution of a noisy action $\boldsymbol{a}_t^{k-1}$ and is well-defined. Additionally, we note that equation 18 is equivalent to (retaining the previous definition for clarity):

$$\max_{\hat{\pi}} \mathbb{E}\left[\sum_{t=0}^{\infty}\gamma^t \sum_{k=0}^{K}\gamma_2^{K-k}\hat{r}(\boldsymbol{s}_t, \boldsymbol{a}_t^k)\right], \tag{26}$$

$$= \max_{\hat{\pi}} \mathbb{E}\left[\sum_{t=0}^{\infty}\left(\prod_{m=0}^{t}\hat{\gamma}_m\right)\hat{r}(\boldsymbol{s}_t, \boldsymbol{a}_t)\right]. \tag{27}$$

$$\tag{28}$$

Thus, we can define the Q function and value function over the new MDP as follows:

$$\hat{Q}(\boldsymbol{s}, \boldsymbol{a}) = \mathbb{E}_{\theta}\left[\sum_{t=0}^{\infty}\left(\prod_{m=0}^{t}\hat{\gamma}_m\right)\hat{r}(\hat{\boldsymbol{s}}_t, \hat{\boldsymbol{a}}_t)|\hat{\boldsymbol{s}}_0 = \boldsymbol{s}, \hat{\boldsymbol{a}}_0 = \boldsymbol{a}\right], \tag{29}$$

$$\hat{V}(\boldsymbol{s}) = \mathbb{E}_{\theta}\left[\sum_{t=0}^{\infty}\left(\prod_{m=0}^{t}\hat{\gamma}_m\right)\hat{r}(\hat{\boldsymbol{s}}_t, \hat{\boldsymbol{a}}_t)|\hat{\boldsymbol{s}}_0 = \boldsymbol{s}\right]. \tag{30}$$

### D.3 DIFFUSION POLICY ON GNMDP

The diffusion policy, as defined in Wang et al. (2022) implicitly parametrizes the policy $\pi$ using a conditional diffusion model $\epsilon_\theta(a^k \mid s)$, wherein the action space $\mathcal{A}$ is conditioned on $\mathcal{S}$. This representation characterizes the distribution of $\boldsymbol{a}_t^{k-1}$ given $\boldsymbol{s}_t, \boldsymbol{a}_t^k$ as follows:

$$\boldsymbol{\mu}_{\theta, s_t}(\boldsymbol{a}_t^k, k) = \frac{1}{\sqrt{\alpha_k}}\left(\boldsymbol{a}_t^k - \frac{1 - \alpha_k}{\sqrt{1 - \bar{\alpha}_k}}\boldsymbol{\epsilon}_{\theta, s_t}(\boldsymbol{a}_t^k, k)\right),$$
$$\boldsymbol{a}_t^{k-1} = \mathcal{N}(\boldsymbol{a}_t^{k-1}; \boldsymbol{\mu}_{\theta, s_t}(\boldsymbol{a}_t^k, k), \boldsymbol{\Sigma}_k), \tag{31}$$

Here, $\boldsymbol{\mu}_{\theta,s_t}$ is the mean of the denoised action based on the noise introduced by $\boldsymbol{\epsilon}_{\theta,s_t}$, and $\boldsymbol{\Sigma}_k$ denotes the covariance matrix. The policy $\pi(\boldsymbol{a}_t \mid \boldsymbol{s}_t)$ involves sequential denoising steps and ultimately yields $\boldsymbol{a}_t = \boldsymbol{a}_t^0$ derived from $\boldsymbol{a}_t^K \sim \mathcal{N}$ given the current state $\boldsymbol{s}_t$.

Given the framework of the GNMDP as defined earlier, it's evident that we can **explicitly** parameterize the $\hat{\pi}$ using the expression equation 31. This parameterization precisely illustrates the generation of $\hat{\boldsymbol{a}}_{I(t,k)} = \boldsymbol{a}_t^{k-1}$ from $\hat{\boldsymbol{s}}_{I(t,k)} = (\boldsymbol{s}_t, \boldsymbol{a}_t^k)$. More formally, this can be articulated as:

$$\boldsymbol{a}_t^{k-1} = \mathcal{N}\left(\boldsymbol{a}_t^{k-1}; \frac{1}{\sqrt{\alpha_k}}\left(\boldsymbol{a}_t^k - \frac{1-\alpha_k}{\sqrt{1-\bar{\alpha}_k}}\boldsymbol{\epsilon}_{\theta,s_t}(\boldsymbol{a}_t^k, k)\right), \boldsymbol{\Sigma}_k\right). \tag{32}$$

This parameterization within the context of the GNMDP structure aligns with the diffusion policy concept and provides a clear mapping between denoised actions in diffusion policy and states and actions in GNMDP.

