# OpenReview forum: "DreamFuser: Value-guided Diffusion Policy for Offline Reinforcement Learning"
_ICLR.cc/2024/Conference — Submitted to ICLR 2024_

### Official Review · Reviewer_6WsC · 2023-10-26

**Soundness:** 3 good
**Presentation:** 3 good
**Contribution:** 2 fair
**Rating:** 6
**Confidence:** 3

**Summary:**

This paper presents Dreamfuser, a diffusion model based offline RL algorithm. Dreamfuser works by jointly optimizing actions for multiple states in parallel, with the diffusion refinement steps integrated into the MDP as "hidden transitions" in the state-action space to make Q-learning tractable in parallel with behavioral cloning.

**Strengths:**

The method presented is algorithmically sound and supported by a good set of comparisons to multiple prior methods to validate its performance. The topic is important and the algorithm novel, to my knowledge.

While I found the writing a little jargon/notation dense at times, overall the paper does a good job of presenting the method.

**Weaknesses:**

While it's a bit of a dirty word, I can't help but feel that this paper is a bit Incremental.

To be clear, I'm of the opinion that that shouldn't be an obstacle to publication- I'm recommending marginal acceptance as I do think the method is novel and sound and the improvement on prior work seems credible, which should be the bar for research to be considered worth sharing.

My general criticism is that this paper doesn't do a lot to justify that the presented algorithm is radically better than prior ones. The motivation makes sense as an extension of previous work, and the experimental results substantiate that Dreamfuser is comparable or better in several benchmarks, but as someone not working in offline RL specifically it's not clear to me how this work has moved the field forward beyond taking that incremental step. Is there an extension or new ground that could be broken to build off of this work?

I've got a few specific questions and suggestions below, but my main concern is that the paper doesn't do a good job explaining its own significance. If there's something I'm missing here I am open to revising my score, for what it's worth.

**Questions:**

-There's a number of grammatical errors scattered throughout the paper, particularly after section 1. I'd suggest another editing pass, though this doesn't significantly impede understanding the paper.

-How does the MDP formulation in Section 4.1 differ from a POMDP (with the action history as part of the observation)? Is there a reason it is not described as a particular subtype of POMDP? The hindsight framing M' seems to suggest that it can be treated as one.

-What is the downside to the GNMDP formulation here? It seems like it should make action/Q-value gradients much more noisy and slow learning due to the temporal extension/noisy credit assignment. Are there drawbacks to this approach?

-I understand the concept, but counting down from K to 0 in the forward direction of time for diffusion steps is confusing when present alongside typical "0 to T" timestep notation. I imagine this counting down notation is standard in the diffusion model literature, but from an RL perspective it would improve readability to have both incrementing in the same direction.

-How does multi-action prediction affect the Gym tasks, similar to Figure 2a and 2b? The results in those figures are compelling, but intuitively I'd expect the importance of temporally extended action planning to vary significantly depending on the task, and this ablation seems to be the main motivation for a major element of the algorithm as presented, as I don't see a clear motivation for predicting a whole trajectory at once in the introduction. Perhaps there's some intuition from previous work I'm unfamiliar with, but it's not obvious to me why joint prediction of actions would improve forward action prediction accuracy at test time (outside a fully model-based method with forward dynamics). Does it benefit supervised training stability in some way?

-In figure 2c, how does this overfitting issue without the dynamics model compare to other methods? I had gotten the sense that methods like CQL generally avoid/minimize overfitting (albeit perhaps at a cost in theoretical performance), am I mistaken and most offline RL methods still suffer overfitting late in training?

-Where does the name "Dreamfuser" come from, out of curiosity? I understand the [dif]fuser part, but what about "dream?"

---

> ### Author Response · Authors · 2023-11-20
> **Comment on Weakness**
>
> **Q: The paper doesn't do a good job explaining its own significance.**
>
> **A:** The motivation and strengths of our work are listed as follows:
>
> * We train the diffusion policy over trajectories and use Q-learning to optimize it. This method outperforms previous approaches, showing a future direction for the following reasons:
>     * The diffusion policy has shown expressive ability in representing behavioral distributions. It could be a significant approach to learning distributions from demonstrations in the future.
>     * Trajectory representation retains temporal relations in action and observation sequences. Investigating algorithms over trajectories could further our understanding in the field of trajectory-based offline RL.
>     * Previous methods primarily focused on behavior cloning of the diffusion policy. However, for practical applications in the future, optimizing over non-optimal demonstrations is essential. Therefore, training the diffusion policy with Q-learning is a worthy problem for discussion.
> * The combination of these features is non-trivial. The diffusion model is notorious for its inference speed, and the trajectory representation would make it even more challenging. On the other hand, both the policy evaluation and policy improvement phases would require sampling actions given the current observations, creating a dilemma.
> * We resolve this dilemma in the following ways:
>     * Instead of viewing sampling from the diffusion model as a whole, we could see the process as a sequence of denoising steps from noise to sampled actions. We can consider these denoising steps as parts of the original MDP. The state of the new one would be represented as $(O, A^k)$ pair, with the action being $A^{k-1}$ given $(O, A^k)$. This transition could be informally defined as from $((O, A^k), A^{k-1})$ to $(O, A^{k-1})$. The transition corresponding to the original MDP would be from $((O, A^0), \epsilon)$ to $((O', \epsilon), \epsilon)$, where $O'$ is the next state of the original transition. These are the primary motivations behind GNMDP. The formal discussion can be found in section 4.2 and appendix D.
>     * The benefit of GNMDP lies in the fact that, without theoretical approximation, we could apply the Q-learning algorithm over a finer granularity. Instead of learning the Q-function over state and action, we learn it over state and noisy action. Rather than optimizing through the whole sampling process, we focus on optimization in a denoising step. With this design, we can use multi-step temporal learning over GNMDP to balance the variance of the Q-function and training speed. Specifically, we borrow the idea of TD3+BC and apply it over GNMDP, resulting in DreamFuser.
> * From the above discussion, we hope the significance of our work becomes clearer. In summary:
>     * We combine promising features to design an RL algorithm. This type of RL algorithm, characterized by expressive ability, optimization capability, and trajectory-based nature, could become mainstream in the future.
>     * We resolve the incompatibility of these beneficial features through a finer granular perspective and apply conventional Q-learning algorithms.
>     * The perspective is inspiring, and the experimental results verify the effectiveness of these approaches. GNMDP is general, allowing various RL algorithms to be applied. The approach shows promise, especially in more complex environments and when trajectory size scales up, areas that could be explored in future work.

---

> ### Author Response · Authors · 2023-11-20
> **Comment on Questions**
>
> **Q1:There's a number of grammatical errors scattered throughout the paper.**
>
> **A1:** We have updated our manuscript and fixed the typos.
>
> **Q2: How does the MDP formulation in Section 4.1 differ from a POMDP?**
>
> **A2:** The GNMDP has no relationship with POMDP. we raise the definition of GNMDP so as to make the design of our algorithm more naturally and easier to understand. I think you would understand the GNMDP better if your could pay attention to the notation in the GNMDP. The subscript *t* refers to the timestep in the original MDP while the superscript *m* refers to the timestep of diffusion process. We set *t* with a forward order while *m* with a backward order in align with the fashion in RL and Diffusion Model respectively. So the $A_{t}^{m}$ means the m-th noisy action at the timestep t of MDP. It is not a form of historical action; And we call it an MDP since we could define state as $(O_t, A_{t}^m)$, action as $A_t^m$, so do the transition. The details are discussed in **Sec 4.2**. Besides, the POMDP focus on P(partial) O(observation) rather than the history action sequence.
>
> **Q3: What is the downside to the GNMDP formulation?**
>
> **A3:** The GNMDP could be seen as a perspective to view the diffusion policy over MDP. We seldom talk about the downside of a view. If necessary, GNMDP is more complex and specialized in depicting diffusion policy. We guess you may wonder the limitation of DreamFuser. Your speculation about noisy gradient of Q function is reasonable. We has paid attention to this and design the rollout len as hyperparameters in B.1 of Appendix for trade-off between computation cost and performance. The multi-step temporal learning can reduce the variance of Q-function.
>
> **Q4: Counting down from K to 0 in the forward direction of time for diffusion steps is confusing.**
>
> **A4:** The counting down in forward direction of time follows the convention of Diffusion Model. We would consider changing to counting up in both directions later.
>
> **Q5: How does multi-action prediction affect the Gym tasks, similar to Figure 2a and 2b**
>
> **A5:** In the supplementary materials, we show the normalized scores of hopper-medium-v2 with different action lengths and plots show a similar pattern like Figure 2(a)(b). But notably, the state and action lengths are not the longer the better, since longer lengths also incur the challenges of extracting the temporal relation of states and actions. As shown in our additional experiment, results with state length 4 are not better than that with state length 3. The benefit of action could be attributed to the feature of hopper environment. The interaction with the hopper environment will cease if the agent falls. Thus, temporally consistent actions sequence might help keep agent stable, in support to our experiment results. Similar to the hopper case, the reason for multi-actions is that the actions of demonstrations in the dataset usually have correlation in general. predicting action sequence can better learn the temporal correlation. For example, the actions of human demonstrations between the adjacent timesteps would not change abruptly. We hope the addtional results and the motivation above could help you.
>
> **Q6: how does this overfitting issue without the dynamics model compare to other methods?**
>
> **A6:** We add discussion about the overfitting at the end of **sec 5.3** to make things clearer. We conjecture the problem may be due to the sparsity of low-level control dataset which mainly collected from simulator by ourselves. The timesteps of low-level control datasets do not exceed 5e4 while the D4RL dataset widely used in offlineRL community accounts for 1e6 timesteps. Thus the previous methods and our experiments over  D4RL dataset do not discuss overfitting problem.
>
> **Q7: Where does the name "Dreamfuser" come from?**
>
> **A7:** It bears our sincere **dream** and hope for the idea of this work could be shared and recognized by the community. We dream about the DreamFuser pushing the progress in the field of reinforcement learning.

---

> ### Comment · Reviewer_6WsC · 2023-11-22
> **Response to Rebuttal**
>
> Thanks for your answers and clarifications!
>
> On a second examination, and given your arguments here, I do see a stronger argument for the significance of this paper's contributions as something more than just a combination of prior methods. I do think the significance could be better argued in the introduction, however, as it seems like at least one of the other reviewers was also confused as to the significance of this paper's contributions.
>
> Looking over the revised manuscript, I think that rewriting sections 1 and 2 to add more intuition for the algorithm and to better frame it in contrast to prior work would be a big improvement and better prime the reader to be receptive to the rest of the paper. The new paragraph in Section 2.2 is a good start, but I'd argue that a full rewrite is worth doing since this note on significance feels tacked on at the end of those sections rather than central to the introduction and related work. Re-reading through Sections 1 and 2, I'm still not coming out with a strong sense of how DreamFuser is distinct from prior works using diffusion models or good intuition for why it ought to outperform such prior methods. A stronger introduction and motivation would resolve most of my remaining concerns about this paper.

---

> ### Author Response · Authors · 2023-11-22
> **Summary of refactoring of section 1 and section 2**
>
> Dear Reviewer,
>
> We would like to express our sincere gratitude to you for meticulously reviewing our work and offering invaluable suggestions. Your advice has been instrumental in refining our manuscript. We have diligently reworked our document, with particular attention given to Sections 1 and 2. This restructuring has allowed us to present a more lucid delineation of related works, our motivations, and the contributions of our research.
>
> The revised content is now organized as follows:
>
> - Introduction:
>     - Highlighting promising techniques for future offline RL development: diffusion policy, sequence modeling, and Q-function guided optimization.
>     - Detailing the motivation and intricate design of GNMDP, along with the benefits of this algorithm. We expound on the challenges, the rationale behind the GNMDP structure, and elucidate the advantages of algorithms based on such frameworks.
>     - Introducing the additional pre-trained dynamics model to enhance the input of the diffusion policy.
>     - Articulating the primary contributions arising from our research endeavors.
>
> - Related Work:
>     - Providing an overview of Offline Reinforcement Learning and referring to the application of TD3+BC within the GNMDP framework.
>     - Discussing the strengths and limitations of the diffusion model in Offline RL.
>     - Emphasizing the motivation behind integrating sequence modeling and highlighting the relevance of the diffusion model in this context.
>     - Offering a concise overview of related works that combine the Q-function with the diffusion model, focusing on crucial aspects and challenges related to this integration.
>
> We genuinely hope that these revisions address your concerns and align more closely with your expectations. Your ongoing feedback and insights are immensely valuable to us, and we eagerly anticipate further discussions to enhance the quality of our work. It would be greatly appreciated if you could reconsider and re-evaluate our revised manuscript.
>
> Thank you sincerely for investing your time and consideration into our work.
>
> Best regards,
>
> The Authors

---

### Official Review · Reviewer_vgVx · 2023-10-31

**Soundness:** 2 fair
**Presentation:** 3 good
**Contribution:** 2 fair
**Rating:** 3
**Confidence:** 4

**Summary:**

The paper proposes DreamFuser, an algorithm that synergistically combines diffusion models and Q-learning for offline reinforcement learning. It models action sequences using a conditional diffusion model and incorporates the diffusion process into the MDP as hidden transitions, enabling efficient Q-function optimization. Experiments on D4RL and robot control tasks demonstrate significant improvements over strong baselines.

**Strengths:**

1. The idea of incorporating the diffusion process into the MDP as hidden transitions is creative, and enables more efficient policy learning and optimization.
2. DreamFuser elegantly combines diffusion modeling and Q-learning in a mutually beneficial way. The sequence modeling handles multimodality while Q-learning enables optimization.
3. The empirical results are quite strong, with DreamFuser outperforming state-of-the-art methods on both D4RL and robotic manipulation tasks. The consistent gains are impressive.The ablation studies clearly demonstrate the value addition from Q-learning, sequence modeling, and dynamics modeling in DreamFuser

**Weaknesses:**

1. The paper does not sufficiently analyze the increased computational costs and inference times resulting from modeling longer action sequences. Using longer sequences increases training and memory requirements, as well as inference latency. A quantification of these costs compared to shorter sequence models would be useful to characterize the trade-offs.
2. The optimal sequence length likely varies across different tasks and environments. More ablation studies on a diverse set of tasks could help identify the best sequence lengths for different problem settings. This could make the method more adaptable and provide more concrete recommendations on sequence length selection.The sequence length is a difficult hyperparameter to tune. The paper could explore adaptive methods to automatically adjust sequence lengths during training to reduce the burden of manual tuning for each task.
3. The paper does not provide an in-depth analysis comparing DreamFuser to prior diffusion model methods like DD and Diffuser using similar length action sequences. Clarifying the key differences and improvements would strengthen the claims.
4. The paper lacks results comparing DreamFuser against Diffusion Policy on the D4RL benchmarks for mujoco and antmaze tasks. This is an essential control experiment to demonstrate the effectiveness of DreamFuser over diffusion modeling alone.
5. From my perspective, this work combines elements from prior work like Q-values guide diffusion policy from diffusion policy, and trajectory generation from diffuser. While the approach is technically sound, the fundamental conceptual advance appears incremental. The novelty may be limited and no insight for me without deeper analysis and comparison.

**Questions:**

Please see the weakness above and address my concerns.

---

> ### Author Response · Authors · 2023-11-20
>
> **Q1: The computation cost by the longer sequence.**
>
> **A1:**  We have paid attention to the compute issue. We did not discuss this part in the following reasons.
> * The sequence length in our experiments does not exceed 10, and due to the parallel computation of neural network, the overhead is low.
> * Instead, the main computation cost comes from the iterative denoising step. So we use DDIM in the DreamFuser. And we design the rollout len as hyperparameters in B.1 of Appendix for trade-off between computation cost and performance. The quantification of these costs could be left as future work when more complex environments are investigated.
>
> **Q2: The ablation about sequence length and adaptive length tuning.**
>
> **A2:** Thanks for the suggestion. However, these parts should be left as future work because
> * The hyperparamter tuning on each indivisual environment may cause overfitting to the feature of each environment. The convention of previous work is to use uniform hyperparameter setting across different environments.
> * Hyperparameter tuning is not related to our key idea of this work.
> * The adaptive tuning is highly non-trivial since the input horizon is related to the network structure. It needs adaptive network, which is a problem itself.
>
> **Q3: Comparison with the previous methods like DD and Diffuser.**
>
> **A3:** The comparison with DD and Diffuser is adopted in the related work part, mainly **sec 2.2**. The key difference is that, compared to previous diffusion policy of trajectory, we introduce the Q learning for optimization of the policy, instead of merely learning the behavior distribution. And instead of importing the Q function into the diffusion polilcy directly, we design the GNMDP structure for diffusion model and RL to blend better.
>
> **Q4: The paper lacks results comparing DreamFuser against Diffusion Policy on the D4RL benchmarks**
>
> **A4:** We evaluate the diffusioin policy over D4RL benchmark as shown in the below table. But the result of diffusion policy on d4rl is optional in our work. As we discuss in the **sec 5.1**, the d4rl benchmark is an indicator for optimization ability, while the diffusion policy is an imitation learning policy, orthogonal to optimization. Also, the paper of this baseline did not offer its results on D4RL. We could supplement these results later into our manuscript.
>
> | Environment                            | Normalized Score |
> |----------------------------------------|------------------|
> | halfcheetah-medium-v2                  | 41.7             |
> | hopper-medium-v2                       | 51.1             |
> | walker2d-medium-v2                     | 77.2             |
> | halfcheetah-medium-expert-v2           | 43.5             |
> | hopper-medium-expert-v2                | 53.6             |
> | walker2d-medium-expert-v2              | 77.2             |
> | **Average score of Mujoco**                 | 57.3             |
> | antmaze-umaze-v0                       | 64.0             |
> | antmaze-umaze-diverse-v0               | 60.0             |
> | antmaze-medium-diverse-v0              | 2.0              |
> | antmaze-large-diverse-v0               | 0.0              |
> | **Average score of Antmaze**    | 31.5             |
>
> **Q5: The novelty of the DreamFuser.**
>
> **A5:** Our strengths are as follows
> * We combine the Q learning with diffusion model and takes the sequence learning. You mention the Q-value guide diffusion policy. But the Q-function in the diffusion policy is pretrained so the optimization ability is limited. **As far as we know, we are the first to combine diffusion model with trajectory-based Reinforcement Learning.**
> * The combination is non-trivial. The diffusion model can be used in a black-box way in the actor-critic training, which is more intuitive. Specifically, in the Bellman update of Q-learning, given $O_t$, $A_t$, $R_t$, $D_t$ and $O'_t$, we can use conditional diffusion model to sample $A'_t$ conditional on $O'_t$, and then we have $Q(O'_t, A'_t)$ for Bellman update. As mentioned in the **sec2.2**, this naive way may cause a full diffusion process of sampling action in the policy training. This computation cost is problematic since it is well-known that the inference speed of diffusion model is still a frontier for community of diffusion model. So we give a new perspective to view the diffusion process the MDP as a whole, that is GNMDP in our work. Through the lens of GNMDP, we could apply the actor-critic training in the manner of DreamFuser. The details are discussed in **sec 4** in our work. In summary, **we design GNMDP structure to optimize the diffusion policy over a finer granularity and design the DreamFuser over GNMDP, with Q-function over noisy action for diffusion policy optimization. As far as we know, we are the first to investigate Q-function over noisy action in the actor-critic frame.**

---

> > ### Comment · Reviewer_vgVx · 2023-11-23
> > **Thanks for the response.**
> >
> > 1. I still raise concerns that the responses to Q3 and Q4 were unsatisfactory.  I still haven't seen results from other diffusion-based policies on D4RL. The claim about the difference between optimization and imitation does not make sense. Additionally,  'Also, the paper of this baseline did not offer its results on D4RL.' is not correct. Diffuser and DD report the results on D4RL, and I believe the comparisons on D4RL is vital and very important.
> >
> > 2. Moreover,  there is overclaiming of novelty in Q5,  such as 'As far as we know, we are the first to combine diffusion model with trajectory-based Reinforcement Learning.' and As far as we know, we are the first to investigate Q-function over noisy action in the actor-critic frame.', which is not convincing for me.  For a easy example, 'Contrastive Energy Prediction for Exact Energy-Guided Diffusion Sampling in Offline Reinforcement Learning', ICML 23'  should as an important baseline in this work. For me, the novelty is limited.
> >
> >  I will maintain the original score until  the author clarifies the positioning of this work correctly and compares it with the existing diffusion-based work fairly.

---

> > > ### Author Response · Authors · 2023-11-23
> > > **Response**
> > >
> > > Dear Reviewer,
> > >
> > > We greatly value your ongoing feedback and sincerely acknowledge the concerns you have raised.
> > >
> > > **Q1: I still haven't seen results from other diffusion-based policies on D4RL and the claim about the difference between optimization and imitation does not make sense**
> > >
> > > **A1:** In our work, we present the baselines of Decision Diffuser (DD) and Diffuser. Additionally, we have included the diffusion policy in our baseline. These methods utilize pretrained value functions as guidance. The value functions of these methods are trained on a clean dataset, while their gradients are back-propagated to the noisy action during the sampling process, potentially leading to theoretical bias. However, in our research, the gradient direction is unbiased as it is trained over GNMDP with a state-noisy-action pair as a new state. We trust this explanation clarifies the distinction between imitation and optimization.
> > >
> > > **Q2: 'Also, the paper of this baseline did not offer its results on D4RL.' is not correct.**
> > >
> > > **A2:** We understand the confusion regarding this matter. To clarify, by "this baseline," we refer to the "diffusion policy" from 'Diffusion Policy Visuomotor Policy Learning via Action Diffusion' (2023). We have independently evaluated the results of this "diffusion policy" on the D4RL benchmark for clarity.
> > >
> > > **Q3: the overclaiming of novelty in the reply of Q5**
> > >
> > > **A3:** We believe our claims are justified and appropriately framed within the scope of existing restrictions. Our trajectory-based Reinforcement Learning approach emphasizes the optimization phase in training, a feature not present in earlier diffusion model-based sequence modeling algorithms. The Q-function over noisy action, as mentioned in CEP, is acknowledged, but our focus is on the actor-critic framework. Unlike CEP, which trains the Q function and extracts the policy from the pretrained value function, our approach underscores the iterative optimization capacity and alignment between the actor and critic within this framework.
> > >
> > > **Q4: CEP (Contrastive Energy Prediction for Exact Energy-Guided Diffusion Sampling in Offline Reinforcement Learning) is referred**
> > >
> > > **A4:** We recognize the CEP and SfBC in our paper's survey stage. However, our research primarily addresses trajectory-based reinforcement learning, contrasting the one-step setting focus of CEP. Consequently, we did not include these works as baselines. We acknowledge that CEP exhibits slightly superior performance over D4RL, potentially due to the dataset's simplicity in revealing the advantages of capturing temporal trajectory correlations.
> > >
> > > We earnestly hope that these clarifications address your concerns and align better with your expectations. Your continuous feedback and insights are invaluable to us, and we look forward to further discussions to enhance the quality of our work. We would deeply appreciate your reconsideration and re-evaluation of our revised manuscript.
> > >
> > > Thank you sincerely for dedicating your time and expertise to our work.
> > >
> > > Best regards,
> > >
> > > The Authors

---

### Official Review · Reviewer_pkRP · 2023-11-01

**Soundness:** 2 fair
**Presentation:** 1 poor
**Contribution:** 2 fair
**Rating:** 3
**Confidence:** 3

**Summary:**

The paper develops a learning protocol that integrates reinforcement learning (RL) with diffusion models, jointly training the underlying diffusion with a Q-function. The results are compared against five baselines.

**Strengths:**

Strengths
* The concept of integrating diffusion models with RL is interesting.
* The paper compares the method against five baselines.
* The paper claims that the method outperforms baselines on most of the considered tasks.

**Weaknesses:**

Summary of weaknesses (for details, see below):
* It is not clear how the "inference" of the method works (i.e., having everything trained: how do we execute actions? is $Q$-value used? do we only sample from the diffusion policy? etc.). There is no pseudo-code provided.
* It is not clear what the components of the method are.
* The loss formulas seem not to reconcile with the formalism of GNMD introduced in the paper.
* The experimental section is lacking.

The method:
* Figure 1, the main visual help to understand what is happening, seems to be attached to Section 4.1. However, it is being explained in  Section 4.2, an unfortunate layout choice, which extends the time needed to understand the approach.
* Section 4.2 is confusing
	* It seems that the dynamics works like this:
		* for $k>0$, sampling an action $A_t^{k-1}\sim \pi(\cdot|\hat{s})$, with $\hat{s}=(O_t, A_t^k)$ transitions to $\hat{s}'=(O_t, A_t^{k-1})$.
			* This suggests that the correct formula for the "consistency" loss should be: $$\mathcal L_{consistency}=\left(\gamma_2\mathbb E_{A_t^{k-2}\sim \pi(\cdot|\hat{s}')} [Q_\phi(\hat{s}', A_{t}^{k-2})] - Q_\phi(\hat{s}, A_t^{k-1}) \right)^2.$$

		* for $k=0$, $\hat{s}=(O_t,A_t^0)$, the action is always $\epsilon\sim N(0,I)$, and the resulting the new state is $\hat{s}'=(O_{t+l}, \epsilon)$,
			* This suggests that the correct formula for the MSBE loss should be:
		$$\mathcal L_{MSBE}=\left((\mathcal R_t + \gamma Q_\phi(\hat{s}', A_{t+l}^{K-1})) -Q_\phi(\hat{s}, \epsilon) \right)^2.$$
	* The last term in equation (9) is unclear.
	* $\epsilon_{t,k}$ in equation (10) is not defined.
	* TD3 is mentioned, but the losses do not reflect this (details are mentioned in passing in Appendix B.1, but no formula is provided).
	* There is no pseudo-code for the training protocol in the main paper (it can only be found in the Appendix).
* In the Introduction, it is written that "DreamFuser integrates a learned dynamic model". It is only briefly mentioned in Section 5, paragraph "effectiveness of the learned model". However, Appendices B.2-3 unexpectedly describe GRU, stating that the diffusion policy now inputs history states $O_t$, noisy sequence $A_t^k$, and predicted future states $O_t'^{k}$ (which is not defined). This seems like an important piece of information, and its omission from the main body of the paper creates confusion as to how the method works. Additionally, transformers make their appearance. What also adds to the confusion is the introduction of $K_{inference}=4$ in B.2. which seems to be used as $K$, but it is not (it is chosen "independently of chosen value for K").

Experiments:
* Results are computed for a low number of seeds (equal to 3, information which can only be found in the Appendix). Additionally, no confidence intervals for the reported numbers are reported, which makes it hard to infer the relative performance of the methods (however, for three seeds, CIs might be too noisy to be informative).
* In Section 5.2, it is mentioned that DreamFuser converges to optimal actions. Why?
* In Section 5.2, it is written that the "proposed trajectory learning optimization technique for Q values proves effective notably excelling in tasks labeled as 'umaze'". What is the reason for that? How is it connected with the diffusion process?
* Figures 2(a) and (c) show strange behavior in learning curves, which is not discussed.
* The results in Figure 2(a-b) are only presented for length 1 or 7. What about other values?

Other:
* It would be interesting to investigate whether the method's success is related to specific features of the chosen environments (and if so, what features).
* In equation (15), there should be $\epsilon \sim N(0, I)$.

**Questions:**

See above.

---

> ### Author Response · Authors · 2023-11-20
> **Comment on Summary of weaknesses**
>
> We thank the reviewer for the expertise and valuable suggestions and feel encouraged by the comment.
>
> **Q1: It is not clear how the "inference" of the method works and no pseudo-code provided.**
>
> **A1:**
> * We add a brief summary about the "inference" at the end of **sec4.3** in our revised version. And B.2 in Appendix is clarified to make inference clear.
>     * For your concerns about the way actions executed, the actions predicted are applied in the environment sequentially; In the first part of **Sec4.1**, we formulate the diffusion policy and dipict how the actions are executed formally. $P'(O_t, A_t)=P'(\{o_{t-h+1}, \dots, o_{t}\}, \{a_{t}, a_{t+1}, \dots, a_{t+l-1}\})=\{o_{t+l-h}, \dots, o_{t+l}\}=O_{t+l}$, $R'(O_t, A_t)=\sum_{i=0}^{l-1} r_{t+i}$ or $\sum_{i=0}^{l-1} \gamma^i r_{t+i}$ given discount factor $\gamma$.
>     * Q function is not used in the inference phase and we only sample from diffusion model for your concerns about the role of Q function and where the actions come from.
> * Due to the page limitations, we regret that the pseudo-code was not included in the previous version. **We supplement the pseudo-code in the appendix A of the updated manuscript.** The primary objective of the inference is to sample an action from the diffusion model, given the current observations. Equation (6) illustrates the formal inference process of the diffusion policy. According to the definition, our DreamFuser represents a specific type of diffusion policy, thereby affirming the validity of the formulation. Furthermore, Figure 1 provides a visual representation that elucidates the inference process. Thus, the inference of the DreamFuser is clear in the main part and we left the pseudo-code into the Appendix.

---

> ### Author Response · Authors · 2023-11-20
> **Comment on the method**
>
> **Q1: The layout of the Figure 1 affect the readability of sec 4.2**
>
> **A1:** Thank you for your time. We have duly noted your request. In the revised version, we include a reference to Figure 1 in **Section 4.1** to provide clarity regarding the explanation of the formulation of the diffusion policy. This reference will specifically highlight how the content of Figure 1 aligns with the discussion in **Section 4.1** for a more coherent presentation.
>
> **Q2: the sec 4.2 is confusing and needs clarification because of typos and writing**
>
> **A2:** We are quite sorry about the presentation and our oversight of some typos. We fix the typos and make things more clear(we hope so) in our revised version. We list them as follows.
>
>
>
> **Q2.1: there are typos about the dynamics**
>
> * The part is not much about dynamic, but more about Bellman update between steps in the GNMDP, which is explained in **Q function learning** part of **sec 4.3**
>
>     * consistency loss should be  $L_{consistency}=( \gamma_2 E_{A_t^{k-1}\sim\hat{\pi}_{\theta} (\cdot \mid \hat{s} )}\hat{Q}\_{\phi} (\hat{s}', A_t^{k-1})-\hat{Q}\_{\phi} (\hat{s}, A_t^{k}) )^2$
>
>     * MSBE loss should be
>         $L_{MSBE}= ((R_t+\gamma \hat{Q}\_{\phi}(\hat{s}', \epsilon))- \hat{Q}\_{\phi}(\hat{s}, A_t))^2.$
>
> **Q2.2: The last term in equation (9) is unclear.**
>
> * The equation (9) is designed to define loss Q as well as revealing the relation between DreamFuser and GNMDP. Thus we fix it in our revised version as $L_Q= -\hat{Q}\_{\phi}(\hat{s}\_{t, k}, \hat{a}\_{t, k})=-\hat{Q}\_{\phi}((O_t, A_t^k), A_t^{k-1}).$
>
> **Q2.3: $\epsilon_{t,k}$ in equation (10) is not defined.**
>
> * We add explanation after equation (10).  $\epsilon_{t,k}$ refers to the sampled diffusion noise corresponding to the $I(t, k)$ step of GNMDP. That is the noise sampled in the forward diffusion process of $A_t$ to $k$-th step.
>
> **Q2.4: TD3 is mentioned, but the losses do not reflect this**
>
> * The TD3 is not mentioned in our work. Instead, we briefly mention **TD3+BC**. We borrow the idea of loss term design from TD3+BC. We mention it to give intuitions about the BC loss term in **Sec 4.3**. BC term serves as regularization term in our work similar to TD3+BC.
>
> **Q2.5: No pseudo-code for the training protocol in the main paper**
>
> * The pseudo-code is not provided because of space limit. The training is fully discussed and all losses are explained in details in **sec4.3**.
>
> **Q3: The appearance of transformers in the discussion about GRU and dynamics model is confusing.**
>
> **A3:** In our architecture, the policy employs a diffusion model with a Transformer backbone to generate actions. The inputs to this architecture are a sequence of observations, denoted as $O_t$, and a sequence of actions, $A_t^k$. Through the denoising process, action sequence $A^{k-1}_t$ is sampled. Thus, the Transformer serves as diffusion model backbone, not related to the dynamics model. **Appendix B.4 in updated version for more details.**
>
> **Q4: The $K_{inference}$ in the Appendix B.2 is unclear.**
>
> **A4:** Our policy, being a diffusion model, benefits from the use of Denoising Diffusion Implicit Models (DDIM) to accelerate inference. This is achieved by selecting a smaller number of diffusion steps, denoted as $K_{inference}$ , for the inference phase with an improved denoising schedule. This choice of $K_{inference}$ allows for faster model evaluation without much compromising the quality of the generated actions. **Appendix B.2 in updated version for more details.**
>
> **Q5: The role of dynamics model is unclear and not discussed in the main script.**
>
> **A5:** Thank you for your insightful comments and for highlighting the need for clarity in our paper. **The last paragraph of sec 4.3 and B.3 in the Appendix try to make it clear in the updated manuscript.** The dynamics model is an integral part of our network architecture, it functions independently from the core algorithmic framework of the diffusion model.  Apart from the diffusion policy, our architecture can optionally incorporate a dynamics model, which augments the input of the policy. This dynamics model, with a GRU backbone, is pre-trained to predict future state transitions, represented as $O'_t$. The inputs of the dynamics model include the noisy actions $A^k_t$ , the previous state $O_t$, and the diffusion timestep $k$. The superscript $k$ on $O'^k_t$ indicates the diffusion timestep associated with its corresponding input, $A^k_t$. The integration of the pre-trained dynamics model allows us to enhance the policy's input with the predicted future state $O'^k_t$. In other words, the diffusion policy inputs $O_t$, $k$, $O'^k_t$, $A^k_t$ and outputs $A^{k-1}_t$ when armed with pretrained dynamics model. The model's role could be viewed as a feature enhancer rather than a fundamental component of the algorithm's design.

---

> ### Author Response · Authors · 2023-11-20
> **Comment on Experiments**
>
> **Q1: Results are computed for a low number of seeds**
>
> **A1:** We follow the field convention. The results for the MuJoCo tasks are supplemented to augment our conclusion. We use 10 random seeds for each environment. And for each environment, we evaluate 20 trajectories and compute the mean normalized scores. Then the results are computed across the seeds to obtain the mean and standard variance of the scores. The changes of scores mean are miner and the average score rises from 89.6 to 90.5. The scores are listed below and the raw data is in the supplementary materials.
>
> | Environment                            | Previous Scores | Current Scores |
> |----------------------------------------|------------------|------------------|
> | halfcheetah-medium-v2                  | 51.0             | **52.8**$\pm$0.0 |
> | hopper-medium-v2                       | 94.7            |  93.8 $\pm$3.5 |
> | walker2d-medium-v2                     | 88.5             | 85.9$\pm$ 0.3|
> | halfcheetah-medium-expert-v2           | 92.9             | **94.8** $\pm$ 0.0|
> | hopper-medium-expert-v2                | 102.3            | **106.3** $\pm$ 3.5 |
> | walker2d-medium-expert-v2              | 109.7            | 109.5 $\pm$ 0.0 |
> | **Average score of Mujoco**                 | 89.6           | **90.5** |
>
> **Q2: In Section 5.2, it is mentioned that DreamFuser converges to optimal actions. Why?**
>
> **A2:** We admit that description here may be inappropriate. We revise this claim in our current version as
> > generating better actions based on the non-optimal sequences through Q-function guided optimization process.
>
> **Q3: The reason for better performance in umaze environment.**
>
> **A3:** The dataset of antmaze, including *umaze*, contains many sub-optimal sequences which do not reach the final goal directly. Therefore, only to optimize over the policy distribution of sub-optimal sequences and stitch the sub-optimal sequences, can the evaluated algorithm perform well in this environment. So instead of answering the reason why DreamFuser can work better in the *umaze* environment, it would be better to say that the good performance in this environment proves the optimization ability of the DreamFuser.
>
> **Q4: The relation between the excellent performance and the adoption of diffusion model.**
>
>  **A4:** The optimization is based on the sub-optimal distributions, which are the basis of optimization. As mentioned in related work, the community has recognize the expressive ability of diffusion policy, which means that the diffusion policy can learn the distribution of dataset better. So the diffusion process leads to a better behavior distribution, thus a better optimized distribution.
>
> **Q5: Figures 2(a) and (c) show strange behavior in learning curves, which is not discussed.**
>
> **A5:** The overfitting issue is discussed in the related section. it would be helpful if "strange  behavior" could be explained more and specified here.
>
> **Q6: Sequence length beside 1 and 7 for Figure2(a)(b)**
>
> **A6:** We implement the experiments with the sequence length 1, 3, 5, 7, 9, 11 and the trend that overfitting is alleviated with longer sequence length keeps in general. The results are shown in the supplementary materials. The figure would be updated in the manuscript later.

---

> ### Author Response · Authors · 2023-11-20
> **Comment on Other**
>
> **how does the method's success relate to specific features of the chosen environments.**
>
> **A1:** We have investigated the method's success with the features of environments. The features of environments can verify the strength of our approach. The discussion about antmaze is mentioned above. Another example might be "hopper" in d4rl, which requires policy to be "conservative" and close to the expert distribution, since it will early terminate with slightly "radical" or biased actions.

---

### Author Response · Authors · 2023-11-20
**Revision Summary**

Thank you very much for your valuable remarks and suggestions. We genuinely appreciate the time you took to review our submission. We summarize all the reviewers' concerns into four categories:
1. On the typos and layouts of our works;
2. On the absent explanation or details of some components and claims;
3. On the reasons that our methods can outperform baselines over tested environments;
4. On the meaning of GNMDP and the novelty of our work;

We have carefully revised our manuscript based on your insightful feedback, addressing the identified typos. For the parts you are concerned with, the contents or the section titles are marked in red. Additionally, we aim to provide further clarification on certain points to ensure a better understanding of the core idea and the significance of our work through our updated submission. We hope that our efforts will help you gain a clearer insight into our research. A short summary of the main modifications made:
1. Go through an additional editing pass and fix the typos;
2. Refactor the sec 1 and sec 2, trying to make our motivation and significance clearer;
3. Supplement the role of pre-trained dynamics model in the last paragraph of sec 4.3;
4. For experiments, we supplement the results of more seeds for MuJoCo and adjust the format. The diffusion policy is added as the baseline in D4RL benchmark. We refactor the ablation study of dynamics model and supplement our explanations;
5. Supplement pseudo-code for inference in Appendix A;
6. Refactor the paragraphs in B. We discuss the inference of the DreamFuser and diffusion model in Appendix B.2 and the dynamics model in Appendix B.3.

---

### Meta-Review · Area_Chair_RY4K · 2023-12-14

**Metareview:**

This paper proposes a new RL algorithm DreamFuser, which combine RL with diffusion models via a trajectory-based multistep approach in contrast to the existing single-timestep approach. The author design the DreamFuser based on the Generalized Noisy Action Markov Decision Process (GNMDP) to alleviate computational issue, and tested it on D4RL to show its superior performance. The explanations in many parts of the paper are not very clear. The paper's experiment also misses important comparison with existing diffusion based RL methods. We thus recommend rejection.

**Justification For Why Not Higher Score:**

The paper misses important comparison.

**Justification For Why Not Lower Score:**

N/A

---

### Decision · Program_Chairs · 2024-01-16

Reject